# The Prop1-like homeobox gene *unc-42* specifies the identity of synaptically connected neurons

**Emily G Berghoff[1], Lori Glenwinkel[1], Abhishek Bhattacharya[1], HaoSheng Sun[1], Erdem Varol[2], Nicki Mohammadi[1], Amelia Antone[1], Yi Feng[1], Ken Nguyen[3], Steven J Cook[1], Jordan F Wood[4], Neda Masoudi[1], Cyril C Cros[1], Yasmin H Ramadan[1], Denise M Ferkey[4], David H Hall[3], Oliver Hobert[1]***

[1]Department of Biological Sciences, Columbia University, Howard Hughes Medical Institute, New York, United States; [2]Department of Statistics, Zuckerman Institute, Columbia University, New York, United States; [3]Dominick P. Purpura Department of Neuroscience, Albert Einstein College of Medicine, Bronx, United States; [4]Department of Biological Sciences, University at Buffalo, The State University of New York, Buffalo, United States

**\*For correspondence:**
or38@columbia.edu

**Abstract** Many neuronal identity regulators are expressed in distinct populations of cells in the nervous system, but their function is often analyzed only in specific isolated cellular contexts, thereby potentially leaving overarching themes in gene function undiscovered. We show here that the *Caenorhabditis elegans* Prop1-like homeobox gene *unc-42* is expressed in 15 distinct sensory, inter- and motor neuron classes throughout the entire *C. elegans* nervous system. Strikingly, all 15 neuron classes expressing *unc-42* are synaptically interconnected, prompting us to investigate whether *unc-42* controls the functional properties of this circuit and perhaps also the assembly of these neurons into functional circuitry. We found that *unc-42* defines the routes of communication between these interconnected neurons by controlling the expression of neurotransmitter pathway genes, neurotransmitter receptors, neuropeptides, and neuropeptide receptors. Anatomical analysis of *unc-42* mutant animals reveals defects in axon pathfinding and synaptic connectivity, paralleled by expression defects of molecules involved in axon pathfinding, cell-cell recognition, and synaptic connectivity. We conclude that *unc-42* establishes functional circuitry by acting as a terminal selector of functionally connected neuron types. We identify a number of additional transcription factors that are also expressed in synaptically connected neurons and propose that terminal selectors may also function as 'circuit organizer transcription factors' to control the assembly of functional circuitry throughout the nervous system. We hypothesize that such organizational properties of transcription factors may be reflective of not only ontogenetic, but perhaps also phylogenetic trajectories of neuronal circuit establishment.

## Introduction

Individual gene regulatory factors are usually expressed in multiple cell types of a developing nervous system, yet their function is often only studied in specific cellular contexts. Many examples illustrate this regional bias in understanding gene function. For instance, the function of the mouse Brn3a POU homeobox gene has been extensively studied in some parts of the central nervous system, such as retinal ganglion cells, habenula, or peripheral sensory organs, but Brn3a function remains largely unexplored in other regions where the gene is expressed, including the interpeduncular nucleus or the superior colliculus (reviewed in *Leyva-Díaz et al., 2020*). Similarly, the function of the LIM homeobox Lhx2 has been well studied in some, but not other parts of the

mouse central nervous system (*Chou and Tole, 2019*). While focused analyses of gene function in specific cellular contexts have provided important cell type-specific insights, broader 'meta-themes' in the function of neuronal differentiation genes may have escaped attention.

Even in a nervous system as limited in size as the *Caenorhabditis elegans* nervous system (118 neuron classes), genetic loss-of-function analysis of specific regulatory factors has also often focused on individual genes in specific cellular contexts. This bias often originated from the phenotype by which a gene was retrieved through mutant screens. For example, the *unc-30/Pitx* transcription factor, one of the first neuronal differentiation genes cloned in *C. elegans* (*Jin et al., 1994*), was identified based on *unc*oordinated locomotory defects (*Brenner, 1974*) and has been extensively studied in the context of ventral nerve cord motor neurons (*Cinar et al., 2005*; *Eastman et al., 1999*; *Howell et al., 2015*; *Jin et al., 1994*; *Petersen et al., 2011*; *Shan et al., 2005*; *Yu et al., 2017*). However, *unc-30* is also expressed in a handful of head neurons (*Jin et al., 1994*; *Reilly et al., 2020*), where its function has remained unstudied. Similarly, the function of the *unc-4* homeobox gene, also retrieved by screens for locomotory defects (*Brenner, 1974*), has been extensively studied in the context of the motor system (*Miller and Niemeyer, 1995*; *Miller et al., 1992*; *Schneider et al., 2012*; *Von Stetina et al., 2007*; *Winnier et al., 1999*), but not in the context of several *unc-4* expressing head neurons (*Miller and Niemeyer, 1995*; *Reilly et al., 2020*). Studying regulatory factors only in isolated cellular contexts may leave overarching themes of gene function undiscovered.

We describe here our nervous system-wide analysis of the *unc-42* gene. *unc-42* mutant animals were also isolated in classic genetic screens for *unc*oordinated locomotion (*Brenner, 1974*), and the gene was subsequently found to code for a phylogenetically conserved homeobox gene, homologous to the vertebrate Prop1 homeobox gene (*Baran et al., 1999*). Vertebrate Prop1 has been mostly characterized for its function in pituitary development (*Watkins-Chow and Camper, 1998*), but the protein is also expressed in unidentified cells in the cerebral cortex, where its function has not yet been analyzed (*Sjöstedt et al., 2020*). Previous work has shown that *unc-42* controls the expression of GPCR-type sensory receptors in the ASH amphid sensory neuron and glutamate-gated ion channels in command interneurons (*Baran et al., 1999*; *Brockie et al., 2001*; *Wightman et al., 2005*). Moreover, these command interneurons were also found to display axon pathfinding defects (*Baran et al., 1999*; *Brockie et al., 2001*). More recent analysis of *unc-42* mutants also identified molecular markers that fail to be expressed in the ASH sensory neurons (*Serrano-Saiz et al., 2013*; *Wood and Ferkey, 2019*), the AIB interneurons (*Bhattacharya et al., 2019*), the AVK interneurons (*Wightman et al., 2005*), and the RMD, SMD, RIV and SIB motor neurons (*Pereira et al., 2015*). However, limitations of available reagents left the expression pattern of *unc-42*, as well as a detailed assessment of the effects of loss of *unc-42* on neuronal differentiation in a fragmented state. Using CRISPR/Cas9-mediated genome engineering as well as neuronal landmark reporters, we describe here the complete expression pattern of *unc-42*, revealing novel sites of expression, and find that the gene is expressed in a synaptically interconnected network of 15 distinct neuron classes. Using molecular marker analysis, *gfp*-based neuronal imaging, and electron micrograph reconstruction, we show that loss of *unc-42* has a profound effect on the proper differentiation and assembly of all 15 *unc-42(+)* neuron classes into functional circuitry, with ensuing deleterious consequences for proper locomotory behavior. Prompted by our analysis of *unc-42*, we examined whether other transcription factors also show a biased expression in synaptically connected neurons. We found many examples of such associations suggesting the existence of 'circuit organizer transcription factors' that operate in overlapping sets of interconnected neurons to instruct the assembly of a nervous system. We discuss the evolutionary implications of our findings.

## Results

### Expression pattern of the *gfp*-tagged *unc-42* locus

The expression of the *unc-42* transcription factor was previously analyzed using antibody staining and reporter constructs including 2.6 kb of sequences upstream of the *unc-42* locus (*Baran et al., 1999*). Tentative cellular identifications of sites of expression were provided for many, but not all, expressing cells (*Supplementary file 1*). We revisited this expression data with a set of three distinct reagents: a chromosomally integrated, multi-copy fosmid-based reporter construct in which the

C-terminus of the locus was tagged with *gfp,* as well as two different engineered strains in which we inserted either *gfp* or *TagRFP* at the 3' end of the *unc-42* locus using CRISPR/Cas9 genome engineering (*Figure 1A*). All three reagents showed the same expression pattern, with the only difference being that the *rfp* strain showed a delayed onset of expression in the embryo, likely due to delayed fluorophore maturation.

The availability of landmark strains for individual neuron types, particularly the novel NeuroPAL landmark strain that allows for disambiguation of all 118 neuron classes (*Yemini et al., 2021*), allowed us to determine the complete pattern of *unc-42* expression during larval development and adulthood. This analysis substantially revised and extended the previously reported expression pattern by Baran and colleagues (*Baran et al., 1999*) (comparison is shown in *Supplementary file 1*). Specifically, we found that in the adult nervous system *unc-42* is strongly and consistently expressed in 40 neurons located in the head of the worm that fall into 15 anatomically distinct neuron classes (*Figure 1B, C*, *Supplementary file 1*, *Figure 1—figure supplement 1*). At the first larval stage, expression is detected in the same set of neuron classes as observed in the adult, with the addition of very weak and inconsistent expression in AVJ and SIA that disappears by the adult stage (*Figure 1—figure supplement 1*). In the embryo, expression is first observed in a few neuroblasts before the bean stage. At the bean stage, when most neurons have terminally divided, *unc-42* expression commences and reaches the full complement of *unc-42(+)* cells at the 1.5-fold stage (*Figure 1B*; *Figure 1—figure supplement 1* for quantification of levels). The cellular sites of expression in the embryonic nervous system appear to be the same as observed post-embryonically, a notion further supported by recent scRNA data (*Packer et al., 2019*). No expression is observed outside the nervous system.

## *unc-42*-expressing neurons are unrelated by lineage but are synaptically interconnected

At first sight, the complete set of neurons that express UNC-42 in the mature nervous system do not share obvious commonalities. UNC-42(+) neurons include sensory, inter-, and motor neurons and display distinct neurotransmitter identities (glutamatergic, cholinergic, or peptidergic) (*Figure 1—figure supplement 1*). Moreover, there is no obvious lineage relationship among the UNC-42(+) neurons (*Figure 1*, *Figure 1—figure supplement 2*). The main worm neuropil can be abstracted into a laminar structure with specific clusters ('strata') or neighborhoods of neurons defined by ultrastructural adjacencies of neuron processes (*Brittin et al., 2021*; *Moyle et al., 2021*). Despite all UNC-42 (+) neurons projecting into this neuropil (*Figure 1E*), UNC-42(+) axons are not exclusively part of one stratum (*Brittin et al., 2021*; *Moyle et al., 2021*; *Supplementary file 2*). However, we noted a striking theme of the 40 neurons (15 neuron classes) that express UNC-42(+): they form a network of densely connected neurons (*Figure 1D, E*).

We tested whether this observation can be explained by chance alone, using a previously described approach that analyzed gene expression in relation to synaptic connectivity (*Arnatkeviciūté et al., 2018*). We first calculated the chance that any two neurons from a random set of 40 neurons in the connectome are connected via chemical or electrical synapses. We then compared this to the chance that any two neurons from a set of 40 *unc-42*-expressing neurons are synaptically connected. We found that *unc-42*-expressing neurons indeed are more likely to be synaptically connected to each other than neuron pairs sampled at random from the whole connectome (*Supplementary file 3*).

One potential problem of this methodology is that among a set of randomly chosen 40 neurons several neurons may not be in close enough physical proximity to potentially form synapses. To address this limitation, we utilized a recently introduced unbiased approach, termed network differential gene expression (NDGE) analysis, with an appropriate null model that takes into account membrane contact (*Taylor et al., 2021*) to estimate the significance of *unc-42* expression with synaptic linkage. This approach is a generalization of differential gene expression analysis (*Wang et al., 2019*) that is in widespread use in single-cell RNA-sequencing literature where the gene expression differences between groups of cells are statistically tested. In NDGE, the groups that are compared are synaptically linked versus non-synaptically linked neuron pairs. Paired multiplicative gene expression that assesses 'homomeric' expression of a gene in synaptically linked neurons yields a p-value that denotes the significance of a gene-pair towards promoting or inhibiting synaptic linkage. The statistical significance of this analysis is driven by a carefully generated null distribution where the

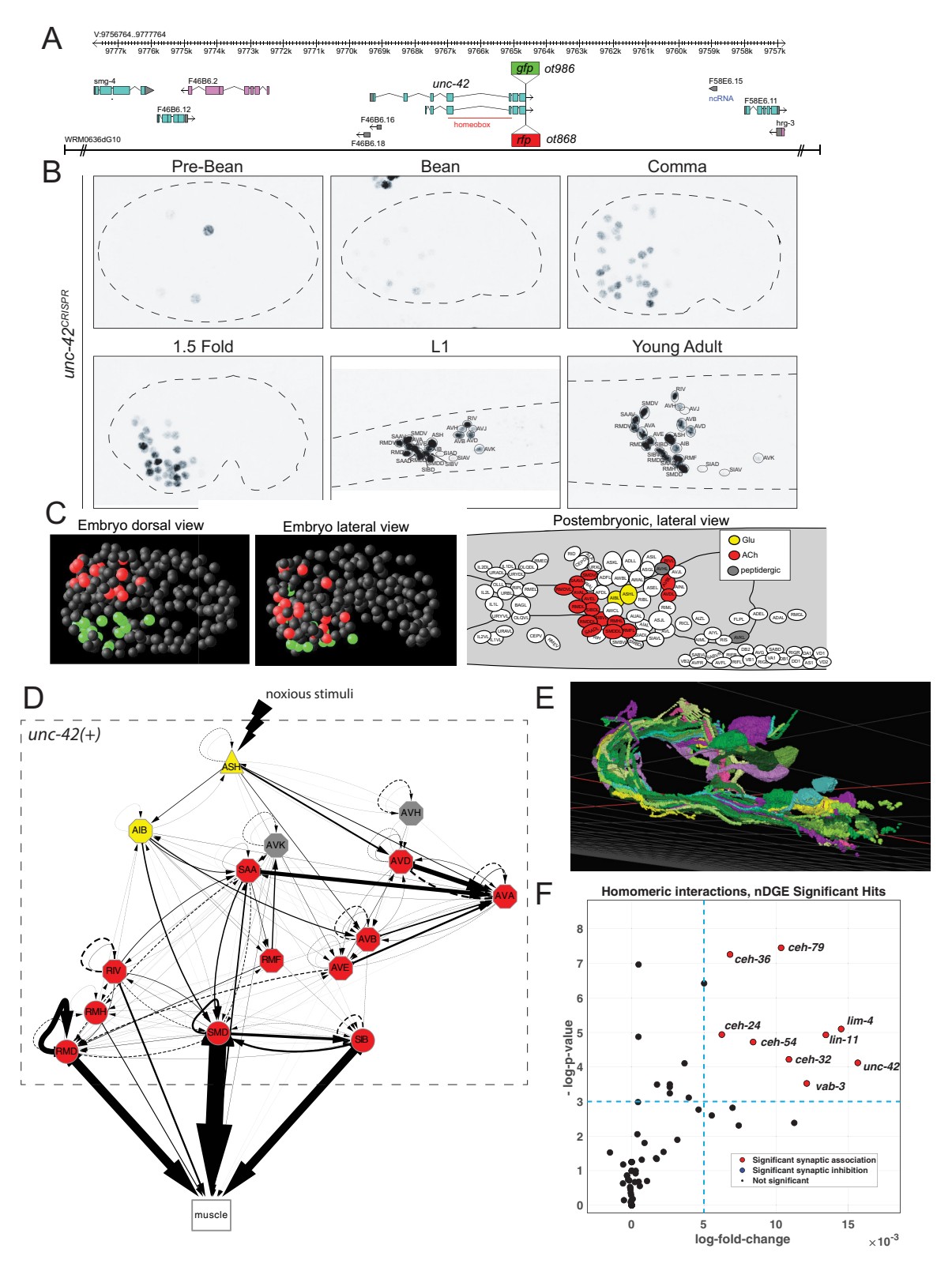

**Figure 1.** *unc-42* is expressed in synaptically connected neurons of a nociceptive reflex circuit. (**A**) *unc-42* reporters used in this study. (**B**) *unc-42* CRISPR-engineered reporter (*ot986*) expression over the course of development. The *unc-42* fosmid reporter shows the same expression in the adult (*Reilly et al., 2020*). Incomplete expression patterns of *unc-42* were previously reported (*Baran et al., 1999*; *Pereira et al., 2015*). See also *Supplementary file 1*. (**C**) Summary of *unc-42*-positive cells in embryonic and postembryonic stages. Color coding in the right panel (lateral view of

*Figure 1 continued on next page*

*Figure 1 continued*

postembryonic stage) indicates neurotransmitter identity, while the color coding in the embryo (middle, left panel) indicates left vs. right bilaterally homologous neurons. (D) Circuit diagram showing all *unc-42*-postive neurons by neuron class (note that only one neuron per class is shown, all neurons are at least bilaterally, in some cases, fourfold-radially symmetric, hence the total of 40 neurons that fall into 15 classes). Color coding is again based on neurotransmitter identity. The display is by Cytoscape (https://cytoscape.org/). Nodes are arranged hierarchically, as described (*Cook et al., 2019*). Node shapes are shown as triangles, sensory neurons; hexagons, interneurons; circles, motor neurons; rectangle, head and neck muscles. Directed edges (arrows) represent chemical synapses. Undirected edges (dashed lines) represent electrical synapses. The width of the edge is proportional to the weight of the connection (the number of serial section electron micrographs where a synaptic specialization is observed). (E) All *unc-42(+)* neurons project to the same neuropil, the nerve ring, where synaptic connections are made. The 3D rendering, which only shows all neurons on the left side of the animal (many of which projecting in the neuropil to the contralateral side of the animal), is based on EM reconstructions and has been generated using at https://www.wormwiring.org (*Cook et al., 2019*). (F) Volcano plot of network differential gene expression (NDGE) analysis on the homeobox gene family, showing the significant homomeric gene interactions associated with synaptic linking (red circles). The X-axis denotes the log-fold gene co-expression difference in synaptically partnered neurons versus non-synaptic neuron pairs. Y-axis shows the negative log-p-value. Red dots indicate gene interactions that have survived the false discovery rate procedure (p<0.05) and log-fold change thresholding (log-FC > 6e-3), while black dots indicate those who did not. Note that log-fold change threshold is lower than traditional thresholding in standard differential gene expression analysis due to the requirement that the selected genes are present in both neurons, a combinatorially rarer event. *unc-42* is indicated to be a significant homomeric gene interaction that is associated with synaptic linking.

The online version of this article includes the following figure supplement(s) for figure 1:

**Figure supplement 1.** *unc-42* expression quantification.

**Figure supplement 2.** *unc-42* expression does not follow lineage history.

connectome is randomly rewired while maintaining its topological properties such as its degree distribution and membrane adjacency (*Rao et al., 1996*). Since many possible combinations of genes could by chance show spurious associations to synaptic linking, we perform false discovery rate (FDR) procedure that limits such false discoveries to 5% (*Benjamini and Hochberg, 1995*). This analysis indeed confirms the significant association of UNC-42 expression with synaptic linkage (p=5.78×10$^{-5}$) (*Figure 1F*).

We further extended these two synaptic association tests to all members of the homeobox gene family that are selectively expressed in a subset of neurons in the mature nervous system (*Reilly et al., 2020*). This analysis shows that the expression of a total of eight homeobox genes each shows a significant enrichment in synaptically connected neurons, both by the NDGE analysis and the methodology by *Arnatkeviciūtė et al., 2018* (*Supplementary file 3*).

## Behavioral consequences of loss of *unc-42*

The set of *unc-42(+)* neurons includes a single sensory neuron involved in nociceptive behavior (*Kaplan and Horvitz, 1993*) and downstream inter- and motor neurons that have been shown, mostly through microsurgical removal or genetic manipulations, to shape the locomotory response to aversive cues, including neurons involved in reversal behavior, omega turns, backward locomotion, restriction of head movement, pausing, and locomotory speed (*Bhattacharya et al., 2019*; *Chalfie et al., 1985*; *Chatzigeorgiou et al., 2013*; *Ezcurra et al., 2011*; *Gray et al., 2005*; *Hamakawa et al., 2015*; *Hart et al., 1999*; *Hart et al., 1995*; *Hilliard et al., 2002*; *Hilliard et al., 2004*; *Kindt et al., 2007*; *Komuniecki et al., 2012*; *Sambongi et al., 1999*; *Shen et al., 2016*; *Walker et al., 2009*; *Yeon et al., 2018*). Using an automated WormTracker system (*Yemini et al., 2013*), we found that animals carrying the canonical allele of *unc-42*, *e419* (a premature stop codon in the homeobox; *Baran et al., 1999*), display behavioral defects that match the defects observed after functional disruption of normally *unc-42(+)* neurons (*Figure 2*; *Supplementary file 4*). This match is not simply the result of *unc-42* mutant animals being completely immobile since there are many locomotory components that are unaffected in *unc-42* mutants (*Supplementary file 4*). Together with the previously reported inability of *unc-42* mutants to respond to aversive sensory cues (*Baran et al., 1999*), this analysis suggests that *unc-42* is essential for *unc-42(+)* neurons to exert their proper function.

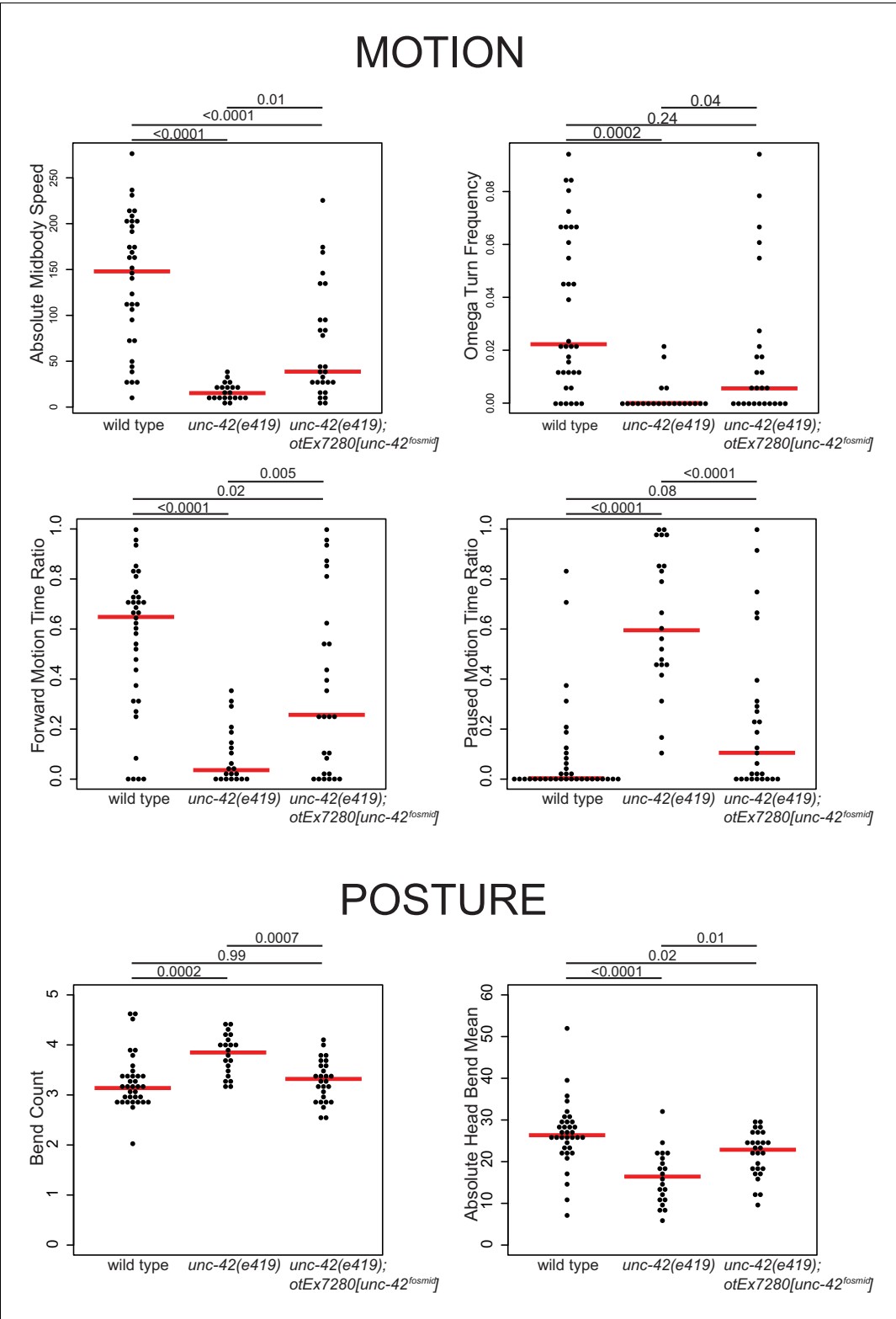

**Figure 2.** Motion and posture defects in *unc-42* mutants. Individual motion and posture features were compared between wild type, *unc-42(e419)*, and *unc-42(e419)* rescue (*unc-42(e419); otEx7280[unc-42^fosmid]*) using the WormTracker. Each circle represents the experimental mean of a single worm. Red lines indicate the median of means. One-way ANOVA followed by Tukey's multiple comparisons test is shown for each comparison. Time ratio = (total time spent performing behavior)/(total assay time).

## *unc-42* is required for cholinergic and glutamatergic synaptic communication

The restriction of *unc-42* expression to a set of synaptically highly interconnected neurons made us first ask whether *unc-42* affects neuronal communication among the *unc-42(+)* neurons, using again the canonical *e419* nonsense allele. To this end, we systematically examined the expression of enzymes and transporters that mark the distinct neurotransmitter identities of the *unc-42(+)* neurons. Some of this analysis had already previously been done for 6 of the 15 *unc-42(+)* neurons: *unc-42* was shown to be required for the glutamatergic identity of the ASH and AIB neurons, as assessed by loss of *eat-4/VGLUT* expression (*Bhattacharya et al., 2019*; *Serrano-Saiz et al., 2013*) and the cholinergic identity of the RMD, SMD, SIB, and RIV motor neurons, as assessed by *unc-17/VAChT* and *cho-1/ChT* expression (*Pereira et al., 2015*). We extended this analysis to the remaining nine cholinergic neuron classes in the *unc-42(+)* circuit. We found that loss of *unc-42* affects *unc-17/VAChT* expression and hence cholinergic identity in all but one of the normally *unc-42(+)* neurons (*Figure 3A–C*). A summary of the effect of *unc-42* on neurotransmitter identity is provided in the context of a circuit diagram that displays all the *unc-42(+)* neurons (*Figure 3D*).

We examined the receiving end of chemical synaptic neurotransmission by analyzing the expression of ionotropic glutamate (Glu) receptors, *nmr-1, glr-1, glr-2, glr-4,* and *glr-5*, and two ionotropic acetylcholine (Ach) receptors, *acr-2* and *acr-15*, in the *unc-42(+)* circuit. The expression and *unc-42* dependence of the ionotropic Glu receptors had been examined before (*Baran et al., 1999*; *Brockie et al., 2001*; *Wightman et al., 2005*). However, we found that some of these receptors are expressed in different set of *unc-42(+)* cells than previously reported likely explained by our usage of updated reagents that more precisely identify neuron classes. We found the expression of each of these genes to be affected in a cell type-specific manner in *unc-42* mutants (*Figure 4*).

Moving from glutamatergic receptors to cholinergic receptors, we found that loss of *unc-42* also affects the expression of cholinergic receptor systems, specifically the AChR subunits *acr-2* and *acr-15* in the RIV, RMD, SAA, and SMD neurons and expression of the AChR-like *des-2* gene in the AVD neurons (*Figure 5A–C*). Moreover, we found that expression of the tyramine-gated chloride channel, LGC-55, which makes the *unc-42(+)* circuit responsive to tyramine signaling from the RIM neurons (*Donnelly et al., 2013*), is affected in the AVB, RMD, and SMD neurons of *unc-42* mutants (*Figure 5D*). A summary of the effect of *unc-42* on neurotransmitter receptor expression is provided in the context of a circuit diagram that displays all the *unc-42(+)* neurons (*Figure 5E*). We note that the effect of *unc-42* on neurotransmitter expression (presynaptic neurotransmitter synthesis/transport and/or postsynaptic neurotransmitter) is often not fully expressive or fully penetrant, indicating that *unc-42* is not the sole determinant of these neuronal identity features. This is a recurrent theme in the analysis of *unc-42* function throughout this paper. We will return to this point at the end of this paper when we discuss transcriptional cofactors of *unc-42*.

## *unc-42* affects peptidergic communication

Extending our analysis beyond chemical synaptic transmission, we asked whether neuromodulatory signaling by neuropeptides is affected within or to/from *unc-42(+)* neurons. Previous work has already shown three neuropeptides (*flp-1, flp-32, nlp-15*) and one neuropeptide receptor (*npr-9*) to be dependent on *unc-42* in specific neuron types (*Bhattacharya et al., 2019*; *Serrano-Saiz et al., 2013*; *Wightman et al., 2005*; *Wood and Ferkey, 2019*). We extended this analysis by examining the expression of seven additional neuropeptide-encoding genes (producing at least 23 distinct neuropeptides) and of eight additional neuropeptide receptors. Several of these peptides and receptors are known cognate ligand/receptor pairs. For example, the *flp-7* and *flp-12* neuropeptides activate the *frpr-8* receptor (I. Beets, pers. comm.) and the *flp-18* neuropeptide binds to the *npr-4 and npr-11* receptors (*Cohen et al., 2009*). We found that the highly patterned expression of these peptides and receptors is severely affected in *unc-42(e419)* mutant animals (*Figure 6A–O*). We again summarize these findings in a circuit diagram (*Figure 6P*), and we again note the often partial penetrance of the effects, as mentioned in the previous section.

Many of these *unc-42*-dependent neuropeptides and neuropeptide receptors have been previously shown to be involved in locomotory behavior, and we found that these behavioral defects match subsets of the defects observed in *unc-42* mutants. For example, animals lacking *flp-18, nlp-15, npr-9,* and *npr-11* display decreased backwards motion (i.e., decreased reversals)

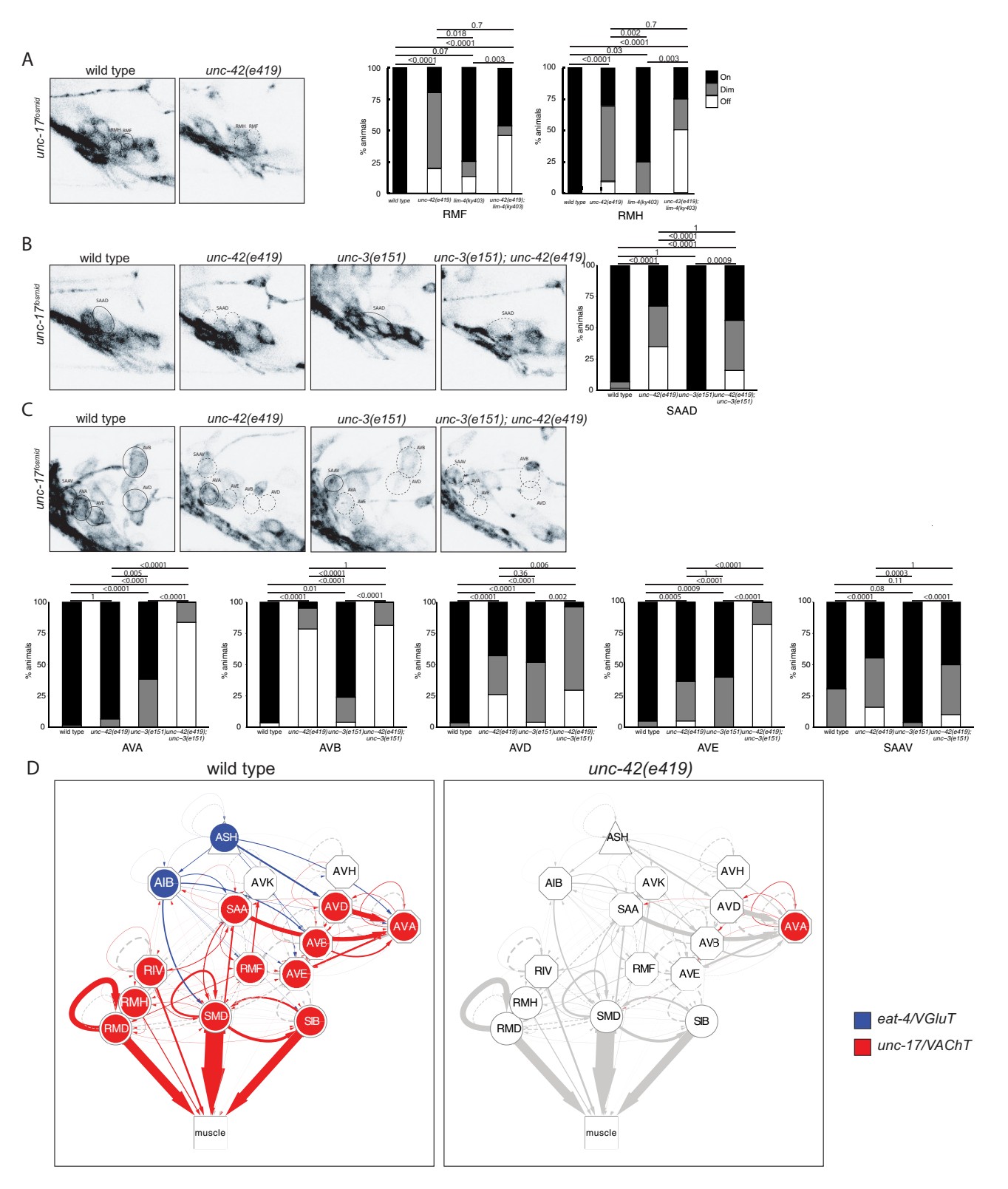

**Figure 3.** *unc-42* affects neurotransmitter identity. (**A**) The expression of the *unc-17/VAChT* fosmid reporter in RMF and RMH neurons is mildly affected in *unc-42* and *lim-4* single mutants, but enhanced in *unc-42; lim-4* double mutants. (**B, C**) The expression of the *unc-17* fosmid reporter in *unc-42* and *unc-3* single mutants and in *unc-42; unc-3* double mutants. (**A–C**) A solid circle indicates expression, and a dashed circle indicates absence of expression. p-values shown by Fisher's exact test. n = 62 wild type, 12 *unc-42(e419)*, 14 *lim-4(ky403)*, 14 *unc-42(e419); lim-4(ky403)*, 26 *unc-3(e151)*, and

Figure 3 continued

32 *unc-42(e419)*; *unc-3(e151)* animals. (**D**) Circuit diagram summarizing the effect of *unc-42* on neurotransmitter identity. *eat-4* data is from **Serrano-Saiz et al., 2013** and **Bhattacharya et al., 2019**. *unc-17* data in SIB, SMD, RIV, and RMD is from **Pereira et al., 2015**. The display is by Cytoscape (https://cytoscape.org/). Nodes are colored to illustrate *eat-4* (blue) and *unc-17* (red) expression. Nodes lose coloring when *eat-4* or *unc-17* expression is affected in an *unc-42* mutant. Edges are colored if the source neuron expresses either *eat-4* or *unc-17*. Edges lose coloring when *eat-4* or *unc-17* expression is affected in the source neuron in *unc-42* mutants (irrespective of whether those effects are partial effects or not). However, note that in this and ensuing circuit diagrams, the existence of gray edges does not indicate whether those edges are generated properly in *unc-42* mutants. Nodes are arranged hierarchically, as described (**Cook et al., 2019**). Node shapes are shown as triangles, sensory neurons; hexagons, interneurons; circles, motor neurons; rectangle, head and neck muscles. Directed edges (arrows) represent chemical synapses. Undirected edges (dashed lines) represent electrical synapses. The width of the edge is proportional to the weight of the connection (the number of serial section electron micrographs where a synaptic specialization is observed).

(**Bhardwaj et al., 2018**; **Campbell et al., 2016**; **Chalasani et al., 2010**; **Yemini et al., 2013**), thereby phenocopying the *unc-42* defects that we have described here (**Figure 2**). Among these, *flp-18* and *npr-11* are expressed in command interneurons that are responsible for backward locomotion, and we found their expression to be *unc-42* dependent in these neurons (**Figure 6C, K**). *npr-11* is also expressed in the AIB interneuron, and, like AIB-ablated animals (**Gray et al., 2005**), animals lacking *npr-11* also phenocopy *unc-42* in that they conduct fewer omega turns. We found that *npr-11* expression in AIB is downregulated in *unc-42* mutants (**Figure 6K**).

In order to further evaluate the potential role of additional neuropeptides and neuropeptide receptors in the locomotory behaviors associated with *unc-42* function, we analyzed the behavior of animals that lack the neuropeptide *flp-21* and the neuropeptide receptors *ntr-1* and *ntr-2*. We found that, like *unc-42* mutants, *flp-21* and *ntr-2* mutant animals displayed decreased backward motion (**Figure 7A**, **Figure 7—figure supplement 1**). *flp-21* is regulated by *unc-42* in the ASH neuron (**Serrano-Saiz et al., 2013**), the sole sensory neuron in the circuit responsible for backwards motion in response to noxious stimuli. Other behaviors in these mutant animals that phenocopied *unc-42* were head bend mean (*ntr-1* and *ntr-2*) and absolute midbody speed (*ntr-1*) (**Figure 7A**). The neurons responsible for these behaviors are yet to be elucidated. Taken together, locomotory defects observed in *unc-42* mutant animals match those observed upon loss of neuropeptidergic signaling systems that are transcriptionally regulated by *unc-42*.

Beyond neuropeptidergic communication, *unc-42* may also affect signaling via other internal signaling systems. We infer this from the observation that *unc-42* also affects the expression of the orphan GPCR *sra-11* in the AVB neurons (**Figure 7—figure supplement 2**). *sra-11* is involved in an associative learning paradigm and responds to an as yet unknown, likely internal signal (**Remy and Hobert, 2005**).

## *unc-42* affects sensory input into the ASH neurons

Direct sensory input into the *unc-42(+)* circuit is provided by the polymodal ASH neurons. Among other modalities, ASH senses high osmolarity (**Kaplan and Horvitz, 1993**) and this sensory paradigm is disrupted in *unc-42* mutants (**Baran et al., 1999**). A putative osmosensor, *osm-10*, is expressed in ASH (**Hart et al., 1999**), and *osm-10* expression is disrupted in *unc-42* mutants (**Figure 7—figure supplement 3**; **Wood and Ferkey, 2019**). ASH also expresses many putative sensory receptors of the G-protein-coupled receptor family, likely involved in the chemorepulsive function of ASH (**Vidal et al., 2018**). The expression of several of these GPCRs, as well as downstream G-alpha proteins, were previously found to require *unc-42* (**Baran et al., 1999**; **Serrano-Saiz et al., 2013**; **Wood and Ferkey, 2019**). We added another GPCR, *srh-15*, to the list of *unc-42*-regulated GPCRs (**Figure 7—figure supplement 3**). Like other transcriptional regulators that control the individuality of distinct sensory neuron types (**Alqadah et al., 2015**; **Masoudi et al., 2018**), we found that *unc-42* does not control expression of the pansensory cilia gene *osm-6* (**Figure 7—figure supplement 3**). In conclusion, *unc-42* controls the proper specification of the sensory neuron that provides sensory input into the *unc-42(+)* nociceptive reflex circuit.

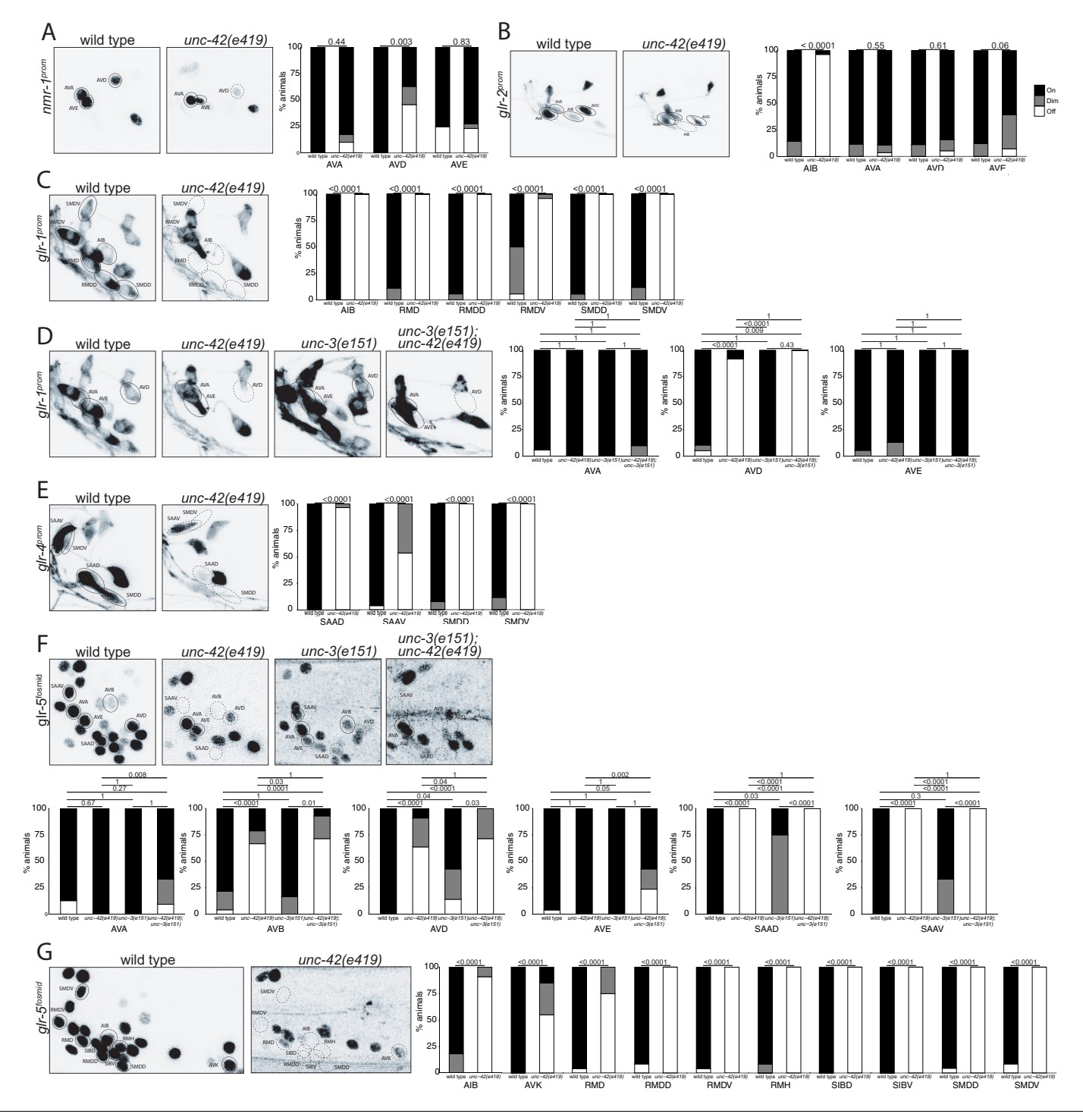

**Figure 4.** *unc-42* affects ionotropic glutamate receptor expression. (**A**) The expression of a *nmr-1* transgene reporter is lost in the AVD neurons in *unc-42* mutants. n = 8 wild type and 69 *unc-42(e419)* animals. (**B**) A *glr-2* reporter transgene shows expression defects in the AIB neurons in the absence of *unc-42*. n = 26 wild type and 28 *unc-42(e419)* animals. (**C**) In the absence of *unc-42*, the AIB, RMD, RMDD, RMDV, SMDD, and SMDV neurons do not show *glr-1* transgene reporter expression. (**D**) The AVD neurons lose expression of the *glr-1* transgene reporter in *unc-42* and *unc-42; unc-3* mutants, but not in *unc-3* mutants. (**C, D**) n = 22 wild type and 24 *unc-42(e419)* animals, 14 *unc-3(e151)*, and 10 *unc-42(e419); unc-3(e151)* animals. (**E**) The expression of a *glr-4* reporter transgene is lost in the SAAD, SAAV, SMDD, and SMDV neurons in *unc-42* mutants. n = 26 wild type and 28 *unc-42(e419)* animals. (**F**) The AVB and SAAV neurons lose expression of the *glr-5* fosmid transgene reporter in *unc-42* and *unc-42; unc-3* mutants, but not in *unc-3* mutants. The expression of the *glr-5* fosmid transgene reporter in the AVD and SAAD neurons is lost in *unc-42*, *unc-3*, and *unc-42; unc-3* mutants. (**G**) In

*Figure 4 continued on next page*

*Figure 4 continued*

the absence of *unc-42*, the AIB, AVK, RMD, RMDD, RMDV, RMH, SIBD, SIBV, SMDD, and SMDV neurons do not show *glr-5* fosmid transgene reporter expression. (F, G) n = 24 wild type, 34 *unc-42(e419)*, and 7 *unc-3(e151)* and 22 *unc-42(e419); unc-3(e151)* animals. (A–G) A solid circle indicates expression, and a dashed circle indicates absence of expression. p-values shown by Fisher's exact test.

## *unc-42* does not affect generation or relative soma position of neurons, but affects axon pathfinding

The data described above demonstrates that *unc-42* affects sensory input into the set of 15 interconnected neurons, as well as communication within these interconnected neurons. We next asked whether apart from controlling neuronal circuit activity *unc-42* may also impact on the assembly of *unc-42(+)* into functional circuitry. We first examined systematically and quantitatively whether neurons that normally express UNC-42 are generated and whether their soma adopt their correct positions in *unc-42* mutant animals. To this end, we again used the NeuroPAL neuronal landmark strain, which labels the position of all neurons with a number of markers, including a panneuronally expressed marker (*Yemini et al., 2021*). We found that all normally *unc-42*-expressing neurons are generated in *unc-42* null mutants, that they express the panneuronal reporter normally, and that they are positioned in a manner that is not quantitatively different from their relative position in wild type animals (*Figure 8A*, *Figure 8—figure supplement 1*).

Previous analysis has described that some command interneurons display axon pathfinding defects along the ventral nerve cord in *unc-42* mutants (*Baran et al., 1999*). Since command interneurons are only a small subset of neurons that express *unc-42,* we sought to expand this analysis to other *unc-42(+)* neurons. This proved to be a substantial challenge because, as we described above (and further below), the vast majority of molecular markers that label individual *unc-42(+)* neurons fail to be expressed in *unc-42* mutant animals. Two exceptions include the *srd-10::gfp* transgene that labels ASH axodendritic morphology and the *hlh-34::gfp* transgene that labels the morphology of the AVH interneuron. Since the expression of both reporters is either not affected or only mildly affected in *unc-42* mutants, they allowed us to visualize ASH and AVH process outgrowth, revealing that *unc-42* is indeed required for proper axon extension of both neuron classes; *unc-42* is required for proper AVH axon extension along the ventral nerve cord, and *unc-42* is required for ASH axons to reach the dorsal midline (*Figure 8B, C*).

We also noted that loss of *unc-42* affects the proper axon extension to the dorsal midline of a subset of other sensory neurons that normally do not express *unc-42* (OLL, AWA, AWB) (*Figure 8E, F*). This apparent cell non-autonomous effect is selective; for example, the ASI neurons are not affected in *unc-42* mutants (*Figure 8—figure supplement 2A*). The apparent non-autonomy of *unc-42* function is further exemplified in the ventral nerve cord, where previous work has revealed axon outgrowth defects in *unc-42* mutants in the HSN, PVQ, and PVP neurons (*Wightman et al., 1997*), none of which normally express *unc-42* (*Figure 1*). As we will show below, *unc-42* affects the expression of the secreted *unc-6/Netrin* axon guidance cue in command interneurons, which may provide a straightforward explanation for the non-autonomous function of *unc-42*.

## *unc-42* mutants display defects in chemical synaptic connectivity

The effect of *unc-42* on axon outgrowth already suggests a role for *unc-42* in the proper assembly of the nociceptive circuit marked by *unc-42* expression. To take this analysis one step further, we asked whether *unc-42* may also affect the generation of chemical synapses in this circuit. To this end, we made use of the availability of *unc-42* mutant animals that were fixed and embedded by Nichol Thomson in the context of early efforts by the Brenner lab to systematically analyze behavioral mutants by electron microscopy (*Brenner, 1973*). For this analysis, the *unc-42(e270)* allele was used, which contains a missense mutation in a highly conserved residue of the homeodomain (*Baran et al., 1999*). We sectioned these blocks and traced all of the axonal process in the nerve ring area (~15 µm) of a single animal. We chose the nerve ring because this is where the majority of synaptic connections between *unc-42(+)* neurons are made. Tracing processes through 309 sections, we were able to assign a total of 19 processes to specific neuron types, 8 of which normally express *unc-42* (see Materials and methods).

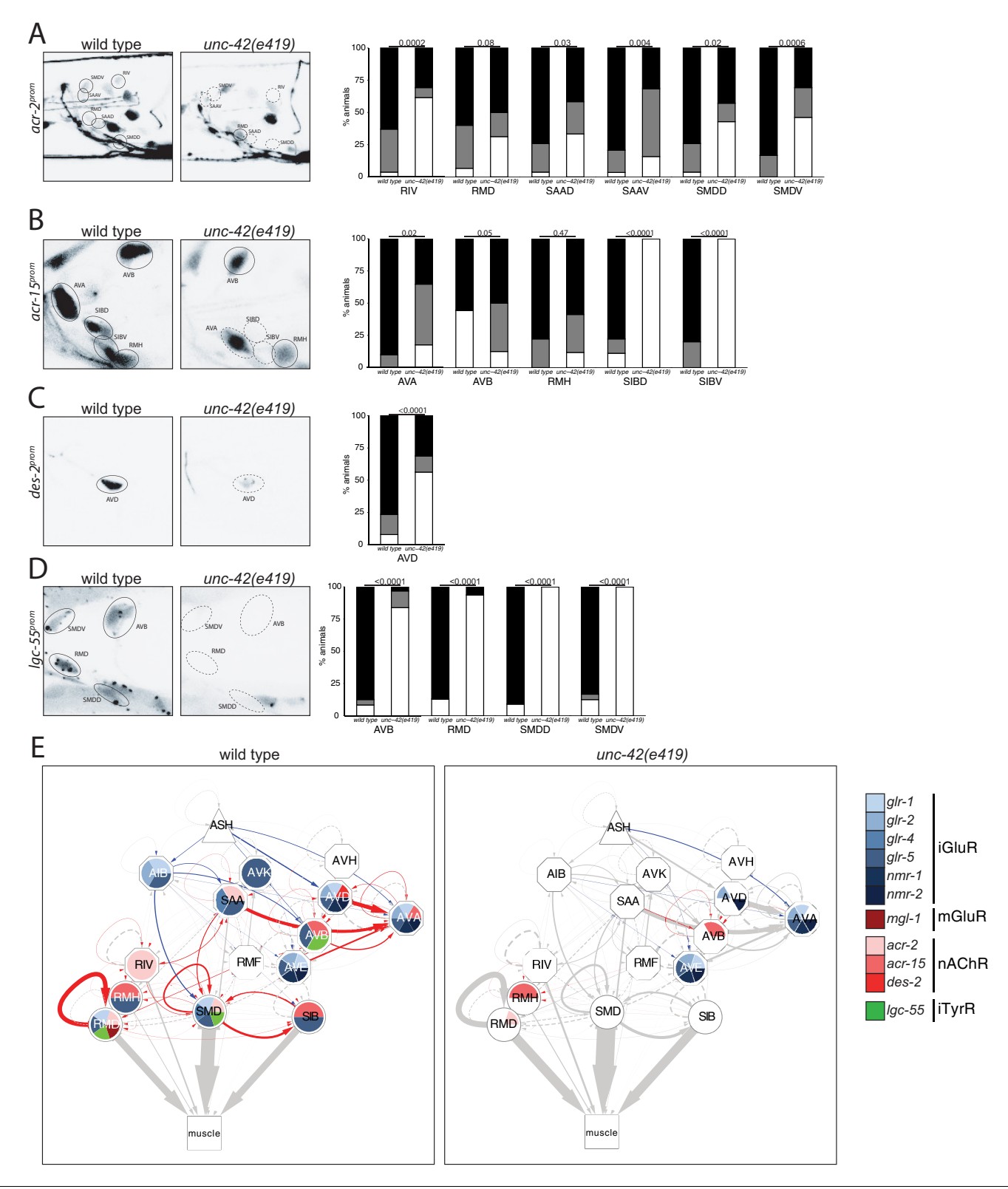

**Figure 5.** *unc-42* affects ionotropic acetylcholine and tyramine receptor expression. (**A**) In the absence of *unc-42*, the RIV, SAAD, SAAV, SMDD, and SMDV neurons do not show *acr-2* transgene reporter expression. n = 30 wild type and 20 *unc-42(e419)* animals. (**B**) An *acr-15* reporter transgene shows expression defects in the AVA, SIBD, and SIBV neurons in the absence of *unc-42*. n = 10 wild type and 18 *unc-42(e419)* animals. (**C**) The expression of a *des-2* transgene reporter is affected in the AVD neurons in *unc-42* mutants. n = 64 wild type and 64 *unc-42(e419)* animals. (**D**) A *lgc-55* reporter

*Figure 5 continued on next page*

*Figure 5 continued*
transgene shows expression defects in the AVB, RMD, SMDD, and SMDV neurons in the absence of *unc-42*. n = 26 wild type and 37 *unc-42(e419)* animals. (**A–D**) A solid circle indicates expression, and a dashed circle indicates absence of expression. p-values shown by Fisher's exact test. (**E**) Circuit diagram summarizing the effect of *unc-42* on neurotransmitter receptor expression as shown in *Figure 4*. *nmr-2* data is from *Brockie et al., 2001*, and *mgl-1* data is from *Pereira et al., 2015*. See legend to *Figure 3* for more information on features of circuit diagram. Edges are colored when the source neuron expresses either *eat-4* or *unc-17* and the target neuron has the appropriate neurotransmitter receptor (see *Figure 3*).

We found that loss of *unc-42* does not affect the overall relative arrangement of the 19 identified neuronal processes with the part of the nerve ring that we were able to analyze (*Figure 9A, B*). We quantified this arrangement through counting sections in which two identified neuron pairs are adjacent to one another and found that the processes that are adjacent to one another in wild type animals retain their adjacency in *unc-42* mutants (*Figure 9C, D*). In terms of synaptic connectivity, we did not observe synaptic defects among neurons that normally do not express *unc-42* (*Figure 9C, D*). However, we found that chemical synaptic connectivity is reduced in neurons that normally express *unc-42* in the pre- and postsynaptic neurons (*Figure 9C, D*). The most striking example of an affected connection in which both neurons are *unc-42(+)* is between the interneurons SAAVL and AVAL. For this connection, the wild type animal makes 22 synapses across 50 serial sections and the *unc-42* mutant animal makes only 2 synapses across two serial sections in spite of the normal adjacency of the two processes. In addition to connections where both the pre- and postsynaptic neurons are normally *unc-42(+)*, some of the connections where only the postsynaptic neuron is normally *unc-42(+)* are also disrupted. For example, in the connection between the presynaptic *unc-42(-)* sensory neuron OLLR and the postsynaptic *unc-42(+)* motor neuron SMDVL, the wild type animal makes five synapses in a total of 11 sections, whereas the *unc-42* mutant animal makes only one synapse across two sections. The above-mentioned process adjacency data indicates that the synaptic defects that we observed cannot be explained by loss of adjacency alone. For example, the wild type SAAVL and AVAL processes are adjacent for 158 sections, whereas in the *unc-42* mutant animal these two neurons are adjacent for 131 sections, a reduction much milder than the loss of chemical synaptic connectivity (22 synapses over 50 serial sections vs. 2 synapses across to 2 serial sections in the *unc-42* mutant). In conclusion, our fine-grained anatomical analysis reveals that while axons are placed correctly in the analyzed *unc-42* mutant animal, there are defects in the generation of synaptic contact. This is consistent with the hypothesis that *unc-42* is required for proper circuit assembly.

## *unc-42* affects electrical synaptic wiring

With the limitations of sample size of the electron micrographical analysis in mind, we pursued alternative means to assess synaptic connectivity in *unc-42* mutants with greater quantitative rigor. Due to the widespread effect of *unc-42* on neuron-class-specific molecular markers, we could not easily utilize GRASP technology to label chemical synapses. However, we were able to assess the integrity of electrical synapses in *unc-42* mutant animals. Electrical synaptic contacts are abundant among the *unc-42(+)* neurons (*Figure 1D*; *Cook et al., 2019*; *White et al., 1986*) and are formed by members of the innexin gap junction proteins (*Hall, 2017*). We had previously mapped the expression pattern of all neuronally expressed innexins (*Bhattacharya et al., 2019*). Several of them are indeed expressed in the *unc-42(+)* nociceptive circuit. *inx-18a* and *inx-19* show a particularly good match, and we examined the expression pattern in *unc-42* mutant animals. We found that in *unc-42* mutant animals *inx-18a* and *inx-19* expression is selectively downregulated in those neurons that normally express *unc-42*. Specifically, *inx-19* expression is affected in ASH sensory neuron, AVA, AVB, AVD, and AVE command interneurons, in the peptidergic AVK interneurons, and the RMD head motor neurons (*Figure 10B*), while *inx-18a* expression is affected in the AVK interneurons, the AVA, AVB, and AVD command interneurons, and the RIV inter/motor neurons (*Figure 10C*). Expression of the very broadly expressed innexin *unc-7* is also affected in a number of neurons in *unc-42* mutant animals (*Figure 10D*). Lastly, in previous studies on the function and regulation of the innexin *inx-6*, we had already shown that *unc-42* affects *inx-6* expression in the AIB neurons (*Bhattacharya et al., 2019*).

To test whether loss of innexins phenocopies the loss of *unc-42,* we tracked *inx-18, inx-19,* and *unc-7* mutant animals and found that several components of locomotory behavior were affected in

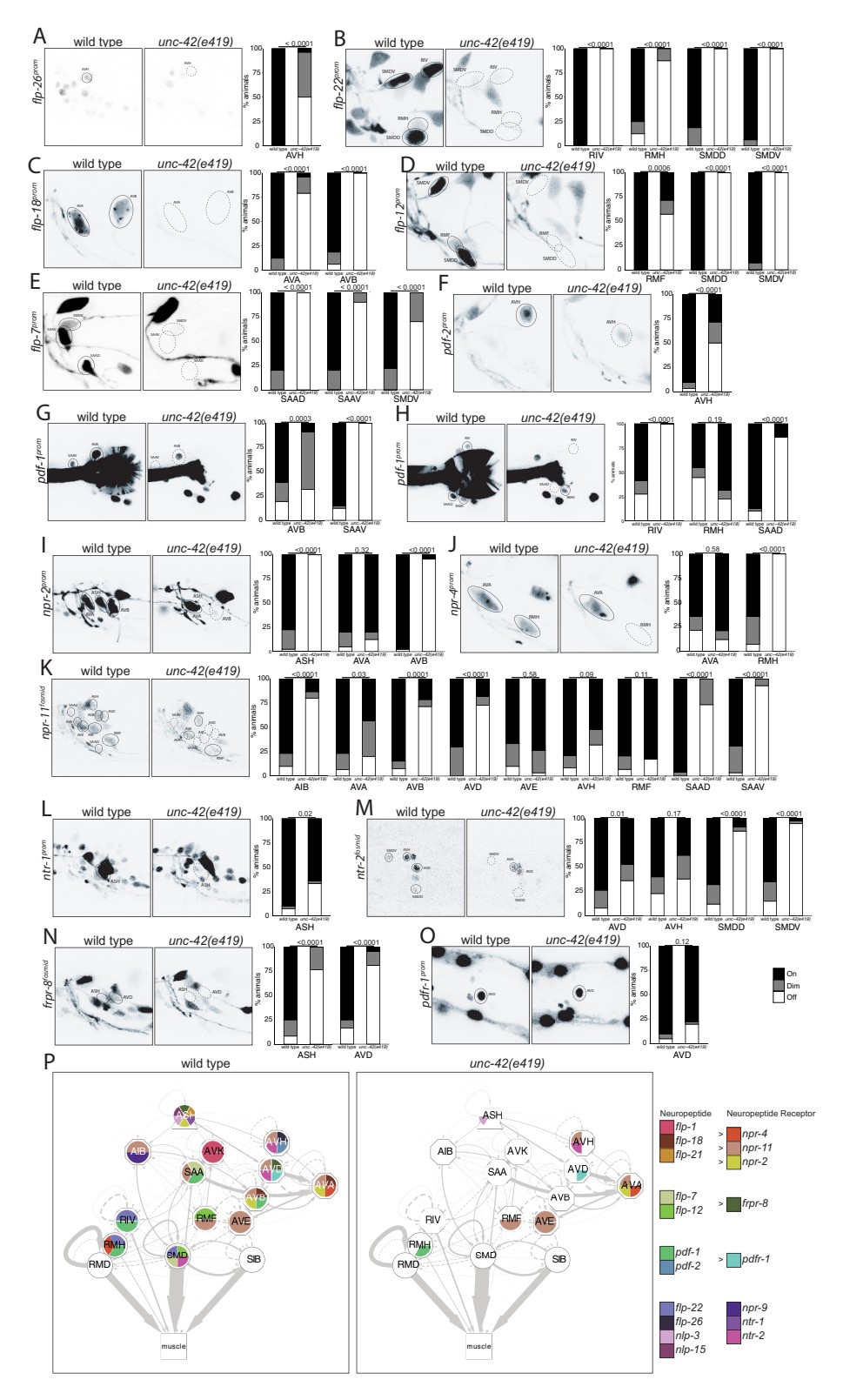

**Figure 6.** *unc-42* affects peptidergic communication . (**A**) A *flp-26* reporter transgene shows expression defects in the AVH neurons in the absence of *unc-42*. n = 22 wild type and 26 *unc-42(e419)* animals. (**B**) A *flp-22* reporter transgene shows expression defects in the RIV, RMH, SMDD, and SMDV neurons in the absence of *unc-42*. n = 16 wild type and 20 *unc-42(e419)* animals. (**C**) The expression of a *flp-18* transgene reporter is affected in the AVA and AVB neurons in *unc-42* mutants. n = 32 wild type and 24 *unc-42(e419)* animals. (**D**) In the absence of *unc-42*, the RMF, SMDD, and SMDV neurons
*Figure 6 continued on next page*

Figure 6 continued

do not show *flp-12* transgene reporter expression. n = 14 wild type and 16 *unc-42(e419)* animals. (**E**) In the absence of *unc-42*, the SAAD, SAAV, and SMDV neurons do not show *flp-7* transgene reporter expression. n = 10 wild type and 20 *unc-42(e419)* animals. (**F**) The AVH neurons lose expression of a *pdf-2* reporter transgene in *unc-42* mutants. n = 32 wild type and 14 *unc-42(e419)* animals. (**G, H**) A *pdf-1* transgene reporter loses expression in AVB, SAAV, RIV, RMH, and SAAD neurons in the absence of *unc-42*. n = 44 wild type and 34 *unc-42(e419)* animals. (**I**) A *npr-2* reporter transgene shows expression defects in the ASH and AVB neurons in the absence of *unc-42*. n = 40 wild type and 40 *unc-42(e419)* animals. (**J**) The expression of a *npr-4* reporter transgene is lost in RMH neurons in *unc-42* mutants. n = 14 wild type and 34 *unc-42(e419)* animals. (**K**) An *npr-11* fosmid transgene reporter shows expression defects in the AIB, AVA, AVB, AVD, SAAD, and SAAV neurons in the absence of *unc-42*. n = 30 wild type and 30 *unc-42(e419)* animals. (**L**) In the absence of *unc-42*, the ASH neurons do not show *ntr-1* transgene reporter expression. n = 40 wild type and 39 *unc-42(e419)* animals. (**M**) The AVD, SMDD, and SMDV neurons lose expression of a *ntr-2* fosmid transgene reporter in *unc-42* mutants. n = 40 wild type and 40 *unc-42(e419)* animals. (**N**) In the absence of *unc-42*, the ASH and AVD neurons do not show *frpr-8* fosmid transgene reporter expression. n = 24 wild type and 24 *unc-42(e419)* animals. (**O**) The expression of a *pdfr-1* reporter transgene in the AVD neurons is unaffected in an *unc-42* mutant. n = 40 wild type and 40 *unc-42(e419)* animals. (**A–O**) A solid circle indicates expression, and a dashed circle indicates absence of expression. p-values shown by Fisher's exact test. (**P**) Circuit diagram summarizing the effect of *unc-42* on neuropeptide and neuropeptide receptor expression. *flp-1* data is from **Wightman et al., 2005**, *flp-21* data is from **Serrano-Saiz et al., 2013**, *nlp-15* and *nlp-3* data is in *unc-42(gk598)* mutants and from **Wood and Ferkey, 2019**, and *npr-9* data is from **Bhattacharya et al., 2019**. See legend to **Figure 3** for more information on features of circuit diagram. Nodes lose coloring when neuropeptide and neuropeptide receptor expression is affected in an *unc-42* mutant.

*inx-19* and *unc-7* mutant animals in a manner that is similar to *unc-42* mutant animals (**Figure 7A**, **Figure 7—figure supplement 1**). Taken together, the prominent loss of expression of innexins is a strong indication that electrical synaptic connectivity is not properly established in *unc-42* mutants.

## *unc-42* affects the expression of the *unc-6*/Netrin guidance cue, of synaptic organizer molecules, and of other cell recognition molecules

Considering the wiring defects of *unc-42* mutants, we asked whether *unc-42* may affect the expression of genes with known or potential roles in cell/cell recognition. The most obvious candidate is the UNC-6/Netrin protein, which is known to affect both axon pathfinding and synapse formation in *C. elegans* and other systems (**Colón-Ramos et al., 2007**; **Hedgecock et al., 1990**). UNC-6 is expressed in a highly restricted manner during *C. elegans* development, and the few sites of neuronal expression include the command interneurons (**Wadsworth et al., 1996**; **Weinberg et al., 2018**). We found that *unc-6*/Netrin null mutants show defects in amphid sensory neuron axon outgrowth that mimic the effects observed in *unc-42* mutants. For example, the ASH axons display a 40% penetrant axon extension defect in *unc-6(ev400)* null mutants (n = 91), comparable to what is observed in *unc-42* mutants (**Figure 8**). We found that in *unc-42* mutants *unc-6*/Netrin expression (as assessed with an *unc-6* fosmid-based reporter reagent) (**Weinberg et al., 2018**) is eliminated from all command interneurons (**Figure 11A**). We attempted to assess whether *unc-42* axon pathfinding defects can be rescued by force-expressing *unc-6* in an *unc-42* mutant background using the *unc-42*-independent *nmr-1* promoter as driver for *unc-6*. Rescue was not observed, possibly due to the late onset of the *nmr-1* driver expression (**Figure 8—figure supplement 2B**). No earlier, *unc-42*-independent command interneuron-restricted drivers are currently available.

The synaptically connected *unc-42(+)* neurons also express a host of cell surface molecules with potential roles in axon pathfinding and synapse formation. We examined 10 cell surface proteins (nine of them of the IgSF family members) that show expression in subsets of *unc-42(+)* neurons for their dependence on *unc-42*. These include genes previously shown to be involved in axon pathfinding and/or axon fasciculation (*lad-2/L1CAM*, *rig-6/Contactin*, *ncam-1/NCAM*) and/or organizing synaptic structure and/or function (*nlg-1/Neuroligin*, *syg-1/KirreL*, *rig-5/IgLON*, *oig-1*, *rig-3*) (**Hashimoto et al., 2009**; **Howell et al., 2015**; **Kim and Emmons, 2017**; **Maro et al., 2015**; **Schwarz et al., 2009**; **Shen and Bargmann, 2003**; **Wang et al., 2008**), as well as two IgCAMs with presently unknown functions (*rig-1* and *rig-5*). These proteins display a unique combinatorial expression in each *unc-42(+)* neurons (with the exception of AVK and AVH). We found that expression of each one of these 10 genes is profoundly affected in *unc-42* mutant animals (**Figure 11B–J**), summarized in **Figure 11K**.

To assess possible function of these cell surface molecules in the *unc-42*-dependent nociceptive reflex circuit, we tested animals that carry mutations in either of seven of the *unc-42*-dependent cell recognition molecules for locomotory defects. We found that *nlg-1*/Neuroligin and three IgSF members, *rig-6*/Contactin, *ncam-1*/NCAM and *rig-3* phenocopy subsets of the *unc-42* locomotory defects

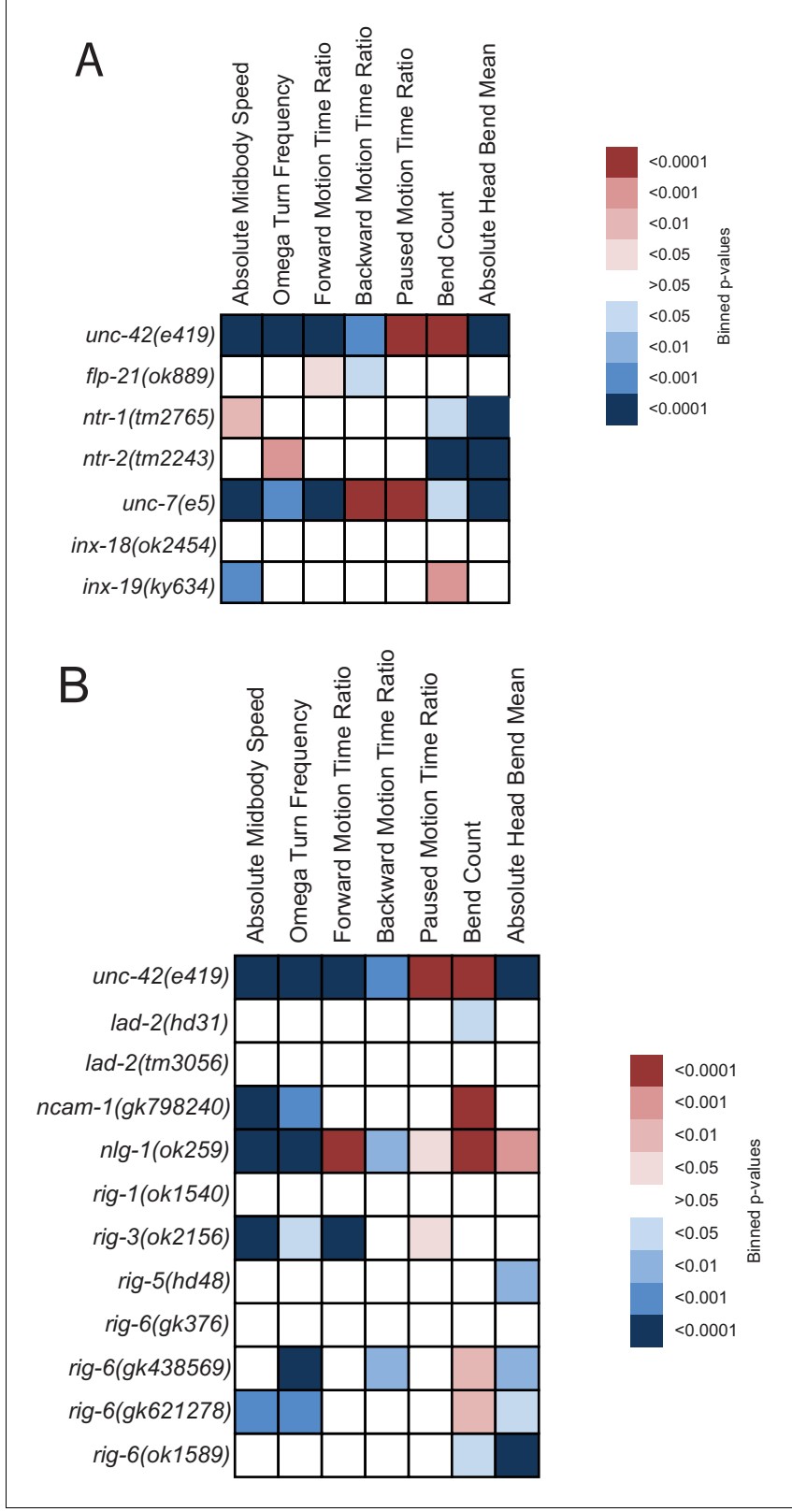

**Figure 7.** Behavioral defects of *unc-42* target genes. (**A, B**) Behavioral phenotypic summaries of the individual motion and posture features identified in *Figure 2* for neuropeptide and neuropeptide receptor mutants (**A**) and for putative cell/cell recognition molecule mutants (**B**). Heat map colors indicate the p-value for each feature for *Figure 7 continued on next page*

*Figure 7 continued*

the comparison between each of the mutant strains and the wild type strain. Red indicates significantly higher p-values while blue indicates significantly lower p-values . See more details in *Figure 7—figure supplement 1*. The online version of this article includes the following figure supplement(s) for figure 7:

**Figure supplement 1.** Loss of neuropeptide, neuropeptide receptors, and putative cell/cell recognition molecules affects locomotion.
**Figure supplement 2.** Effects of *unc-42* on *sra-11* expression.
**Figure supplement 3.** Effects of *unc-42* loss on ASH differentiation.

(*Figure 7B*, *Figure 7—figure supplement 1*). Similar to *unc-42* mutants, *rig-6* and *ncam-1* mutants display fewer omega turns and are expressed, in an *unc-42*-dependent manner, in omega turn controlling neurons (ASH, RIV, or AIB). *ncam-1(-)* animals also phenocopy *unc-42(-)* animals in regard to increased pausing, a behavior that is controlled by the AIB neuron, where the expression of *ncam-1* is *unc-42* dependent. *nlg-1* mutants phenocopy *unc-42* mutants in decreased backwards motion, a behavior that is controlled by the AVA command interneuron, where the expression of *nlg-1* is *unc-42*-dependent (*Figure 7B*). Cases in which we observed no or limited locomotory defects are not conclusive because several of the available mutants are not clear molecular nulls. As with other cases described above where a target gene mutation phenocopies aspects of the *unc-42* mutant phenotype, it is important to realize that the phenocopy does not prove that the respective target gene indeed can be made responsible for the *unc-42* mutant phenotype (e.g., the target gene could affect the phenotype from a complete different cell where *unc-42* displays no function). Nevertheless, the phenocopy is an encouraging prerequisite for being a functionally relevant target of *unc-42.*

We observed no obvious defects in axon pathfinding or synaptic vesicle clustering (*rab-3::gfp* marker) of a subset of *unc-42(+)* neurons (ASH, AIB, SAA) in *rig-6*, *rig-3*, *ncam-1*, and *nlg-1* mutants (*Figure 8—figure supplement 3*), but since RAB-3::GFP labels presynaptic sites indiscriminately, this approach lacks the anatomical resolution to draw any definitive conclusions. More analysis will be required to assess whether these genes may affect synaptic connectivity of *unc-42(+)* neurons.

## Different transcription factors interact with *unc-42* to specify distinct neuronal identities

The analysis described above raises a number of questions. First, does UNC-42 control the many target genes described above directly or indirectly? Second, how does UNC-42 activate distinct target genes in distinct neuron types? Using the single cell transcriptome atlas of all *C. elegans* neurons (*Taylor et al., 2021*) and a phylogenetic footprinting pipeline, described in the accompanying paper by Glenwinkel et al., we found that functionally validated UNC-42 binding sites are enriched in the cellular transcriptomes of each one of the 15 neuron classes that express UNC-42 including those genes whose expression we have shown here to be *unc-42*-dependent (representative examples are shown in *Figure 12A*) (Glenwinkel et al., accompanying paper). This analysis strongly suggests that UNC-42 directly activates the expression of terminal gene batteries in all *unc-42(+)* neurons.

If UNC-42 binding sites are enriched in all the UNC-42-dependent target genes described above, why are they not activated in all UNC-42(+) neurons? The most parsimonious explanation is that UNC-42 requires neuron-type-specific cofactors to activate distinct sets of neuron class-specific gene batteries. We found this scenario to indeed apply in many UNC-42(+) neuron classes. For example, we had previously shown that the EBF/Collier ortholog *unc-3* affects the cholinergic identity of command interneurons in a manner similar to the *unc-42* effect (*Pereira et al., 2015*). Consistent with UNC-3 and UNC-42 working together, we found that UNC-42 binding sites are co-enriched with UNC-3 sites in genes expressed in all command interneurons (*Figure 12B*). To functionally validate this potential interaction, we made use of the fact that *unc-42* and *unc-3* single mutants each only display a partially penetrant loss of *unc-17/VAChT* expression. Building *unc-42; unc-3* double mutant animals, we found that in some of the affected neurons the defects are synergistic relative to the single null mutants, indicating that both genes cooperate to control cholinergic neurotransmitter identity in the command interneurons (*Figure 3*).

Since command interneurons come in different types, one may expect that UNC-42 and UNC-3 interact with distinct additional factors in distinct command interneuron types. Loss of the ARID-type

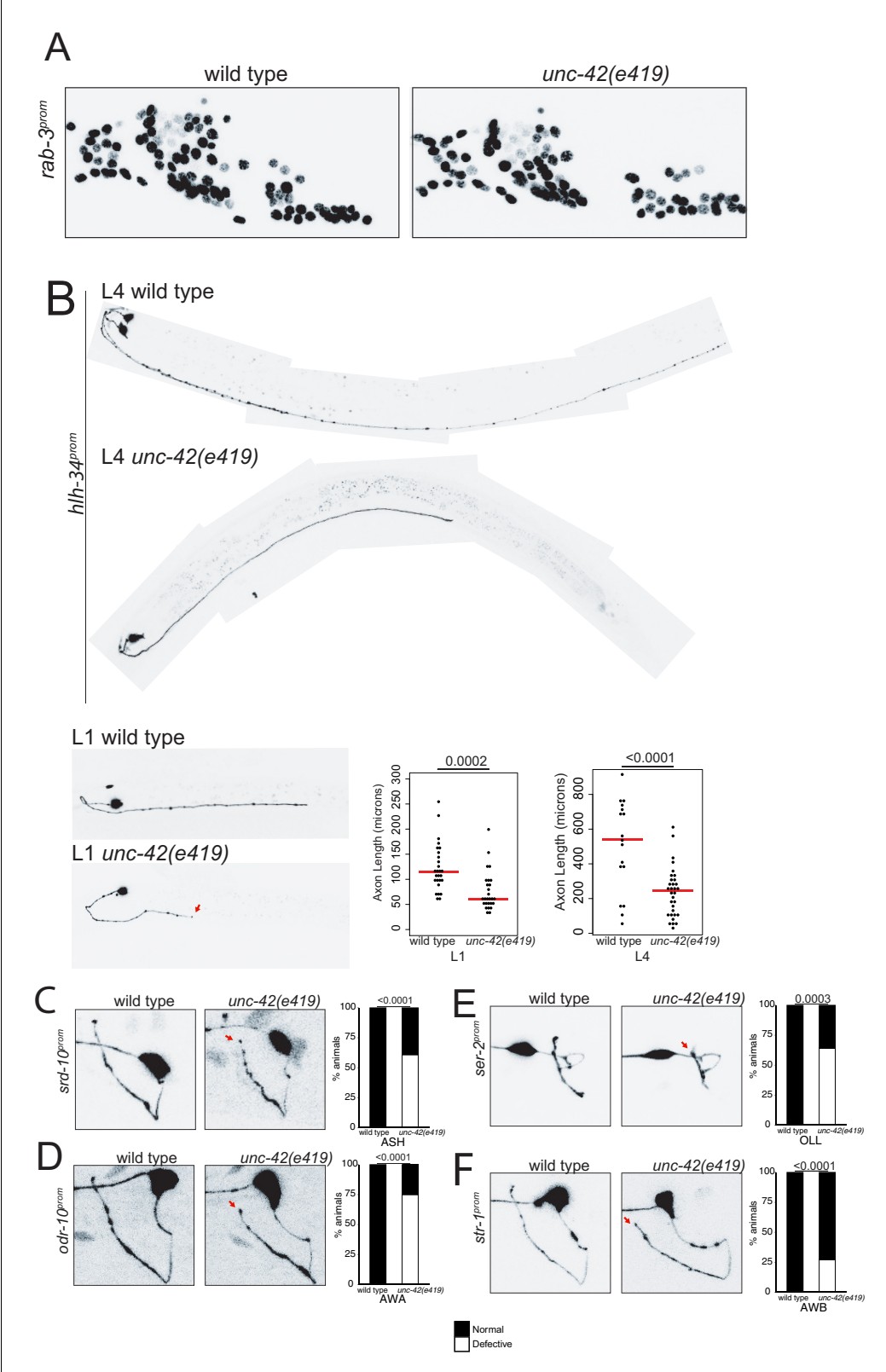

**Figure 8.** *unc-42* does not affect generation of relative soma position of neurons but partially affects axon extension of some neuron classes. (**A**) A *rab-3* pan-neuronal reporter transgene does not show defects in neuron generation and relative soma position in the absence of *unc-42*. See also *Figure 8—figure supplement 1*. (**B**) In the absence of *unc-42*, AVH neurons display axon extension defects in the ventral nerve cord at the L1 and L4 larval stages. Each circle represents one animal. Red lines indicate the median. p-values shown by one-way ANOVA followed by Tukey's multiple

*Figure 8 continued on next page*

*Figure 8 continued*

comparisons test. (**C–F**) ASH, AWA, OLL, and AWB neurons display nerve ring axon extension defects in *unc-42* mutants. Arrow indicates the axon extension defects. p-values shown by Fisher's exact test. (**C**) n = 29 wild type and 46 *unc-42(e419)* animals. (**D**) n = 30 wild type and 32 *unc-42(e419)* animals. (**E**) n = 18 wild type and 14 *unc-42(e419)* animals. (**F**) n = 68 wild type and 70 *unc-42(e419)* animals. See also *Figure 8—figure supplement 3*. The online version of this article includes the following figure supplement(s) for figure 8:

**Figure supplement 1.** *unc-42* does not control cell soma position.
**Figure supplement 2.** Rescue of *unc-42* in the command interneurons does not restore axon anatomy, and loss of *unc-42* does not affect axon anatomy in all neurons.
**Figure supplement 3.** Putative cell/cell recognition molecules and the *unc-6* netrin guidance cue do not affect presynaptic specializations.

*cfi-1* transcription factor causes differentiation defects in one of the command neurons, the AVD neurons (*Shaham and Bargmann, 2002*) resembling those observed in *unc-42* mutants. In contrast, the nuclear receptor *fax-1* controls the expression of genes in AVA and AVE, but not AVD (*Wightman et al., 2005*). Since binding sites for CFI-1 and FAX-1 are also defined, we again used the phylogenetic footprinting pipeline described in the accompanying paper by Glenwinkel et al. We found that the AVD gene battery contains an enrichment for CFI-1 binding sites (in addition to the UNC-42 and UNC-3 binding site enrichment), while the other command interneurons display enrichments of FAX-1 binding sites (in addition to the UNC-42 and UNC-3 binding site enrichment) (*Figure 12B*). Notably, the AVK interneurons that require both *fax-1* and *unc-42* for their proper differentiation (*Wightman et al., 2005*) also display a co-enrichment for FAX-1 and UNC-42 binding sites. Hence, we surmise that UNC-42 cooperates with UNC-3 and CFI-1 to specify AVD identity and with UNC-3 and FAX-1 to specify AVA and AVE identity, and with FAX-1, but not UNC-3, to specify AVK identity.

Apart from synergistic activities of *unc-42* and *unc-3* in command interneurons, we identified another genetic interaction in the RMF and RMH head motor neurons. As described above, *unc-42* single mutants have mild defects in execution of cholinergic fate of these neurons (*unc-17/VAChT* expression) (*Figure 3*). The LIM homeobox gene *lim-4* was previously shown to be expressed in a number of mostly cholinergic head sensory, inter-, and motor neurons and acts as a terminal selector to control AWB and SMB neuron identity (*Sagasti et al., 1999*; *Kim et al., 2015*; *Alqadah et al., 2015*; *Pereira et al., 2015*). However, its function in other neurons was not previously investigated. We find that like *unc-42* single mutants *lim-4* single mutants have mild defects in the execution of cholinergic fate in the RMF and RMH head motor neurons (*unc-17/VAChT* expression) (*Figure 3*). These defects are strongly enhanced in *unc-42; lim-4* double mutant (*Figure 3*).

Moving beyond cholinergic neurons, we considered the previously uncharacterized AVH interneuron, whose identity specification is affected in *unc-42* mutants, as described above. The *hlh-34* gene, a bHLH-PAS transcription factor, is exclusively expressed in the AVH neuron throughout the life of the AVH neuron (see Materials and methods). We used CRISPR/Cas9 to engineer a loss-of-function allele of *hlh-34* (*Figure 13A*) and found AVH differentiation defects in these animals (*Figure 13B, C*) that match those observed in *unc-42* animals. Moreover, *hlh-34; unc-42* double null mutants show strongly enhanced mutant phenotypes (*Figure 13C*). Hence, UNC-42 may cooperate with HLH-34 to specify AVH identity.

In the glutamatergic AIB interneuron, we identified two potential collaborators for *unc-42*, the Meis-type homeobox gene *unc-62* and the Pbx-type homeobox gene *ceh-20*, both of which co-expressed with *unc-42* exclusively in the AIB neurons (*Reilly et al., 2020*). *unc-62/Meis* and *ceh-20/Pbx* mutant animals phenocopy the AIB differentiation defects observed in *unc-42* mutant animals (*Figure 13D–I*). Specifically, loss-of-function alleles of both genes results in defects in expression of several key identity features of AIB, including loss of glutamatergic identity (*eat-4/VGLUT* expression) and loss of expression of several innexin genes (*Figure 13D, E, G*).

Lastly, in the SIB lateral motor neurons, the differentiation defects observed in *unc-42* mutants are phenocopied by loss of the *ceh-24* homeobox gene (*Schwarz and Bringmann, 2017*), suggesting that these two transcription factors may cooperate in the SIB neurons. Because the effect of both mutants is fully penetrant, we refrained from building *unc-42; ceh-24* double mutants. The *unc-42(+)* SMD neurons had also been reported to express *ceh-24* and require *ceh-24* for their correct specification (*Schwarz and Bringmann, 2017*). However, using the cell identification and cell fate tool NeuroPAL, we found that *ceh-24* is expressed in SMB, not SMD (*Reilly et al., 2020*), and,

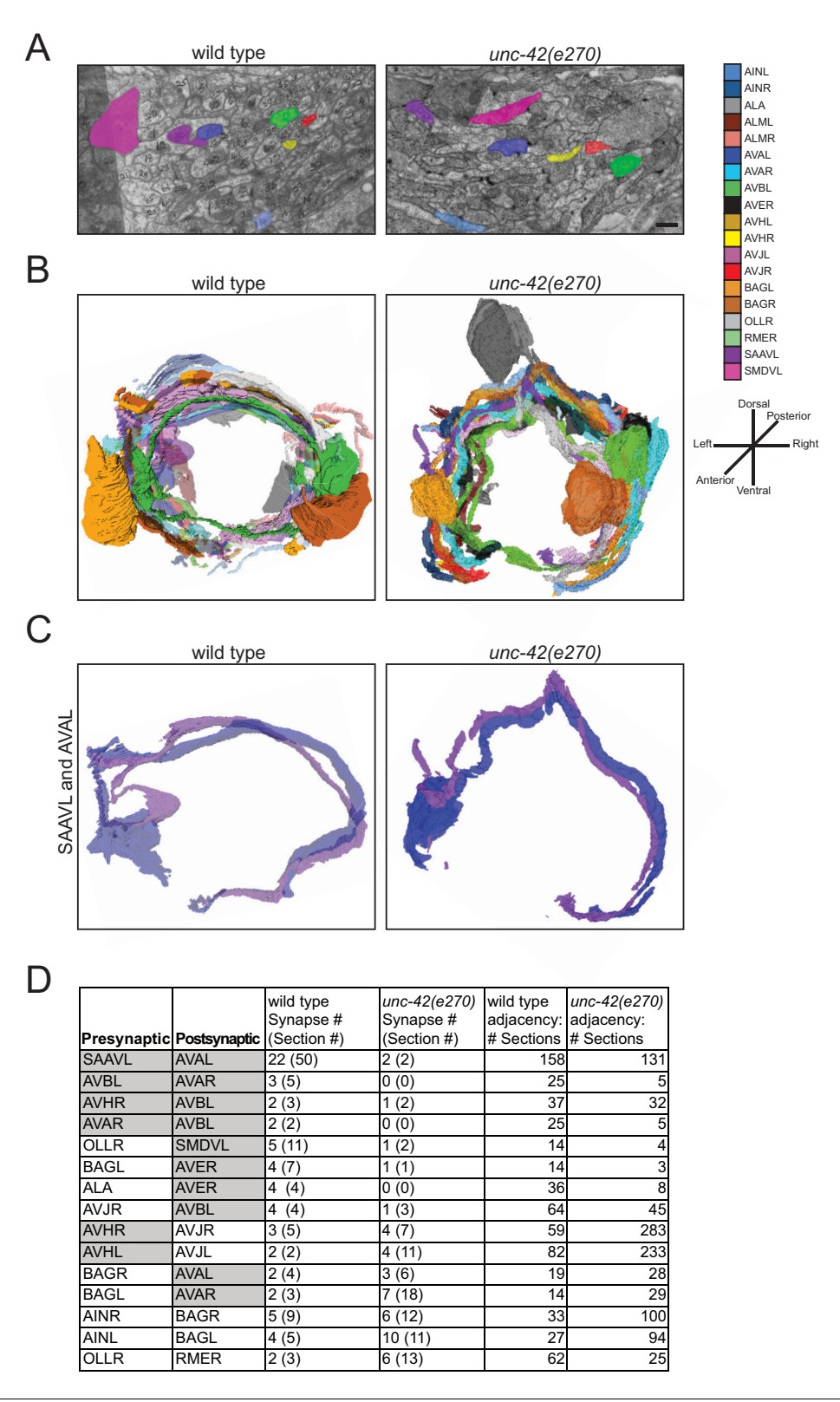

**Figure 9.** *unc-42* affects synaptic connectivity. (**A**) A transverse section of the right ventral ganglion in *wild type* (*N2U*, section 142) is compared to the corresponding section in *unc-42(e270)*. The following processes are shown: AINL, AVAL, AVBL, AVHR, AVJR, SAAVL, and SMDVL. Scale bar, 500 nm. (**B**) Three-dimensional renderings of all identified neurons in *unc-42(e270)* are compared to *wild type* (*N2U*). Transverse view, posterior is to the back. (**C**) *Figure 9 continued on next page*

Three-dimensional renderings of SAAVL and AVAL in *wild type* and *unc-42(e270)*. (D) Tabular summary of synaptic contacts and adjacencies of all identified neurons in *unc-42(e270)* compared to *wild type* (*N2U*). Gray cells are *unc-42*-expressing neurons.

correspondingly, that in *ceh-24* null mutants, SMB and not SMD differentiation is defective (***Figure 13—figure supplement 1***).

As summarized in ***Figure 14***, we conclude that the specificity of UNC-42 activity is determined by cell-type-specific collaboration of UNC-42 with distinct cofactors. Such collaboration could, for example, be in the form of cooperative DNA binding, as previously observed for other Prd-type transcription factors (***Wenick and Hobert, 2004***), or could be in the context of forming a 'transcription factor collective' that operates additively to control target gene expression (***Junion et al., 2012***). Cohorts of collaborating terminal selectors have been termed 'core regulatory complexes' (***Arendt et al., 2016***).

## Feedforward regulation of *unc-42* cofactors by *unc-42*

There is much precedent in the literature for transcription factors operating in the context of feedforward loops in which a given transcription factor activates a downstream transcription factor to then cooperate with the downstream transcription factor to control entire batteries of effector genes. The first described example in the *C. elegans* literature is the UNC-86 POU homeodomain transcription factor, which activates the MEC-3 LIM homeodomain transcription factor to then co-regulate together with MEC-3 the expression of scores of touch neuron receptor genes (***Duggan et al., 1998***; ***Zhang et al., 2002***). We therefore asked whether *unc-42* controls the expression of the transcription factors that appear to cooperate with UNC-42 to control neuronal identity. We indeed find that the expression of *ceh-24*, *cfi-1*, and *unc-3* reporter genes is strongly affected in *unc-42* mutants (***Figure 13—figure supplement 2***). Moreover, *fax-1* has previously been shown to be regulated by *unc-42* (***Wightman et al., 2005***). In contrast, *hlh-34*, *unc-62/Meis*, and *ceh-20/Pbx* expression is unaffected in AVH and AIB, respectively. Together, this data lends support to the notion of the existence of positive feedforward loops in cell identity specification in a number of different contexts. More needs to be learned about these feedforward loops to understand why *unc-42* activates these factors only in some, but not other neurons; possibly, the ability of *unc-42* to control downstream transcription factors depends on the presence of cell-specific lineage cues that are present transiently when the respective neuron is born. A precedent for this is the activation of the terminal selector *ceh-10* by its own cofactor, the terminal selector *ttx-3*, exclusively in the AIY interneuron, which requires transient Wnt signaling cues (***Bertrand and Hobert, 2009***).

## Collaborators of *unc-42* are also expressed in synaptically connected neurons

Having defined the roles of UNC-42 in the neurons in which the protein is expressed, we circled back to our original observation that all the 15 distinct UNC-42(+) neuron classes are more highly interconnected than expected from any random set of 15 neuron classes. Intriguingly, several of the factors that collaborate with *unc-42* in a neuron-type-specific manner are also expressed in synaptically connected neurons. This includes the *ceh-24* NK2-type homeobox gene, which cooperates with *unc-42* in the SAA and SIB neurons (summarized in ***Figure 14***). In addition, *ceh-24* is also expressed in the RME, SIA, and SMB neurons (***Reilly et al., 2020***). All five *ceh-24* expression neurons are interconnected more heavily than expected by chance. This correlation is observed using the analysis as done by Arnatkeviciute et al., as well as the NDGE analysis that we described above (***Supplementary file 1***, ***Figure 1F***). Similarly, the LIM homeobox gene *lim-4* collaborates with *unc-42* in some neurons, but is also expressed in additional sets of synaptically connected neurons (***Supplementary file 1***, ***Figure 1F***). Lastly, the sites of expression of the COE-type transcription factor *unc-3*, an apparent cofactor for *unc-42* in command interneurons (see above), are also significantly enriched for synaptically connected neurons based on the approach by Arnatkeviciute et al. and NDGE analysis. These UNC-3(+) neurons include the UNC-42(+) command interneurons, but also UNC-42(-) ventral cord motor neurons that are directly innervated by command interneurons. Taken together, one can imagine that the *C. elegans* connectome can be deconstructed into a series

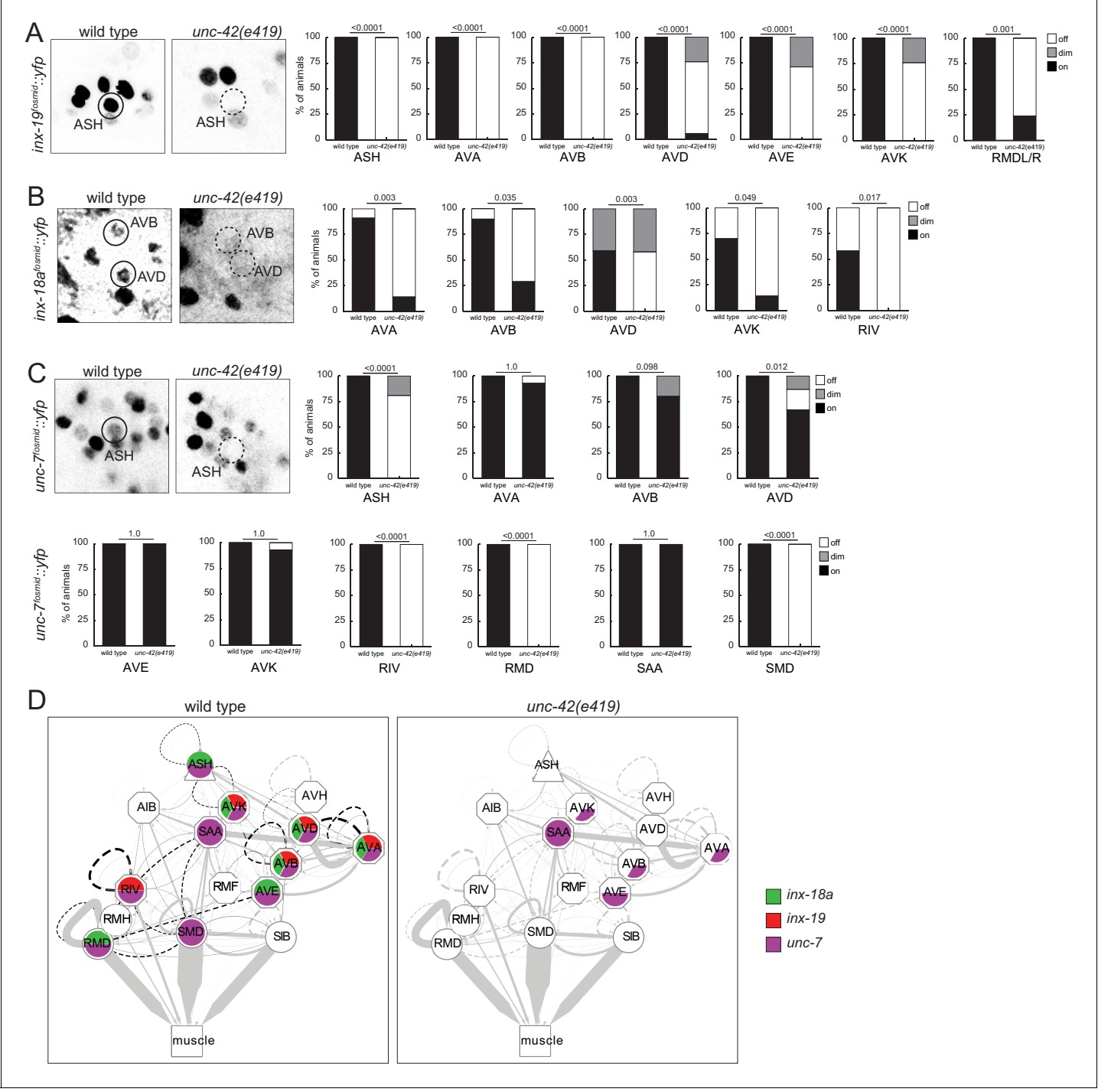

**Figure 10.** *unc-42* affects electrical synaptic communication (innexins). (**A**) A *inx-19* reporter transgene shows expression defects in the ASH, AVA, AVB, AVD, AVE, AVK, and RMDL/R neurons in the absence of *unc-42*. (**B**) A *inx-18a* reporter transgene shows expression defects in the AVA, AVB, AVD, AVK, and RIV neurons in the absence of *unc-42*. (**C**) A *unc-7* reporter transgene shows expression defects in the ASH, RIV, RMD, and SMD neurons in the absence of *unc-42*. Expression of *unc-7* reporter remained unaffected in the AVA, AVB, AVD, AVE, AVK, and SAA neurons in the absence of *unc-42* (cells were identified by relative position). p-values shown by Fisher's exact test. (**D**) Circuit diagram summarizing the effect of *unc-42* on innexin expression. See legend to *Figure 3* for more information on features of circuit diagram. Edges are colored in black to indicate electrical synaptic connections between neurons that were examined in this analysis. Edges lose coloring when *inx-18a, inx*-19, or *unc-7* expression is affected in either neuron in *unc-42* mutants (irrespective of whether those effects are partial effects or not).

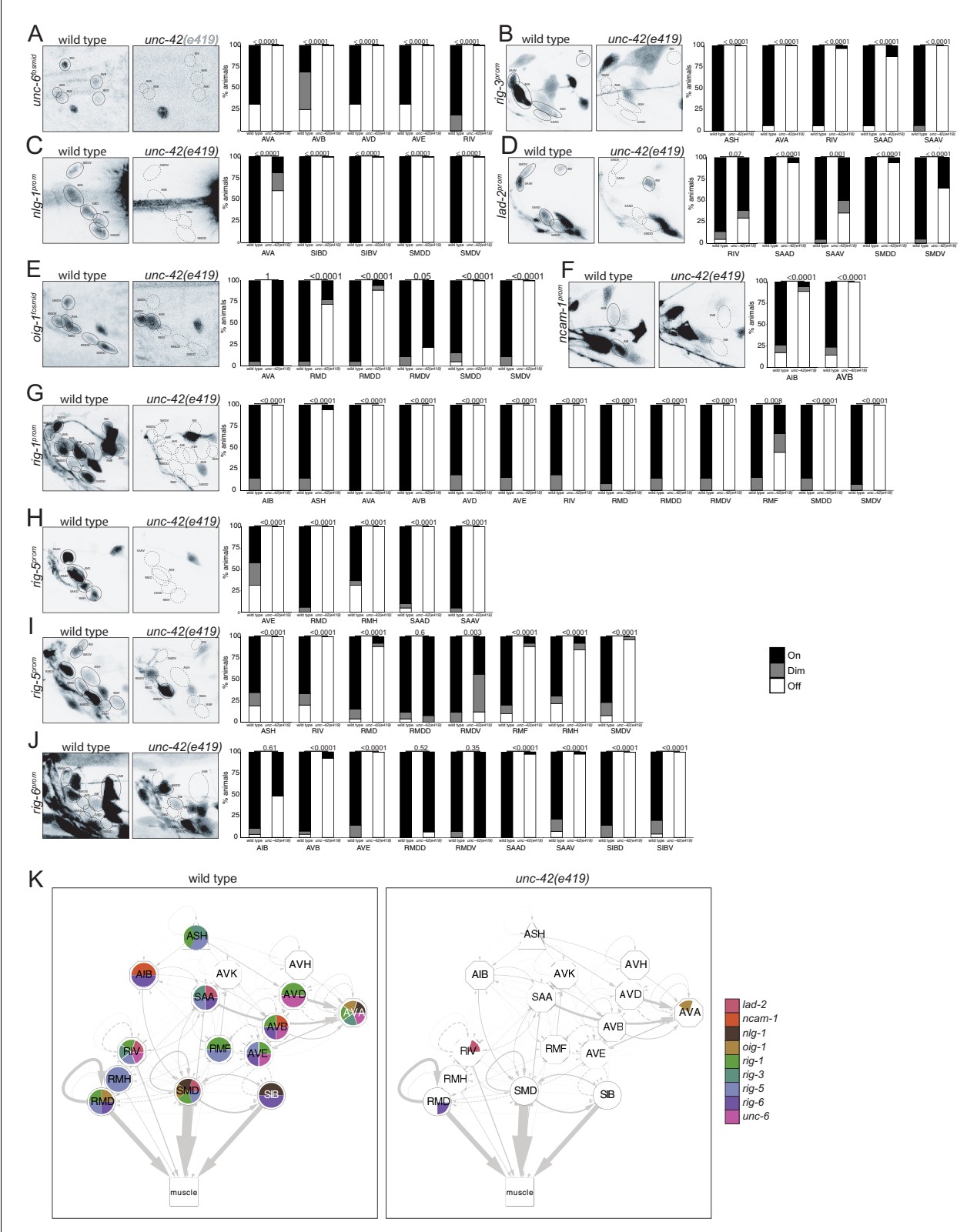

**Figure 11.** *unc-42* affects putative cell/cell recognition molecules (IgSFs) and the *unc-6* netrin guidance cue. (**A**) In the absence of *unc-42*, the AVA, AVB, AVD, AVD, and RIV neurons do not show *unc-6* fosmid transgene reporter expression. n = 16 wild type and 24 *unc-42(e419)* animals. (**B**) A *rig-3* reporter transgene shows expression defects in the ASH, AVA, RIV, SAAD, and SAAV neurons in *unc-42* mutants. n = 15 wild type and 32 *unc-42(e419)* animals. (**C**) A *nlg-1* reporter transgene shows expression defects in the AVA, SIBD, SIBV, SMDD, and SMDV neurons in the absence of *unc-42*. n = 20

*Figure 11 continued on next page*

*Figure 11 continued*

wild type and 38 *unc-42(e419)* animals. (**D**) The SAAD, SAAV, SMDD, and SMDV neurons lose expression of a *lad-2* reporter transgene in *unc-42* mutants. n = 21 wild type and 34 *unc-42(e419)* animals. (**E**) In *unc-42* mutants, the RMD, RMDD, SMDD, and SMDV neurons do not show *oig-1* fosmid transgene reporter expression. n = 16 wild type and 18 *unc-42(e419)* animals. (**F**) A *ncam-1* reporter transgene show expression defects in AIB and AVB in the absence of *unc-42*. n = 24 wild type and 36 *unc-42(e419)* animals. (**G**) A *rig-1* reporter transgene shows expression defects in the AIB, ASH, AVA, AVB, AVD, AVE, RIV, RMD, RMDD, RMDV, RMF, SMDD, and SMDV neurons in the absence of *unc-42*. n = 16 wild type and 18 *unc-42(e419)* animals. (**H**) A *rig-5* reporter transgene (*otEx5883*) show expression defects in AVE, RMD, RMH, SAAD, and SAAV neurons in *unc-42* mutants. n = 26 wild type and 20 *unc-42(e419)* animals. (**I**) The expression of a *rig-5* reporter transgene (*hdEx332*) is lost in the ASH, RIV, RMD, RMDV, RMF, RMH, and SMDV neurons in the absence of *unc-42*. n = 26 wild type and 28 *unc-42(e419)* animals. (**J**) The AVB, AVE, SAAD, SAAV, SIBD, and SIBV neurons lose expression of a *rig-6* reporter transgene in *unc-42* mutants. n = 28 wild type and 52 *unc-42(e419)* animals. (**A–J**) A solid circle indicates expression, and a dashed circle indicates absence of expression. p-values shown by Fisher's exact test. (**K**) Circuit diagram summarizing the effect of *unc-42* on the expression of putative cell/cell recognition molecules and the *unc-6* netrin guidance cue. See legend to **Figure 3** for more information on features of circuit diagram. Nodes lose coloring when the expression of putative cell/cell recognition molecules and the *unc-6* netrin guidance cue is affected in an *unc-42* mutant (irrespective of whether those effects are partial effects or not).

of overlapping groups of interconnected neurons whose interconnectivity is defined by what we propose to call 'circuit organizer transcription factors' (**Figure 14**). These factors operate as both terminal selectors to control molecular identify features of a neuron, such as its neurotransmitter identity, and may also organize neurons into synaptic circuits.

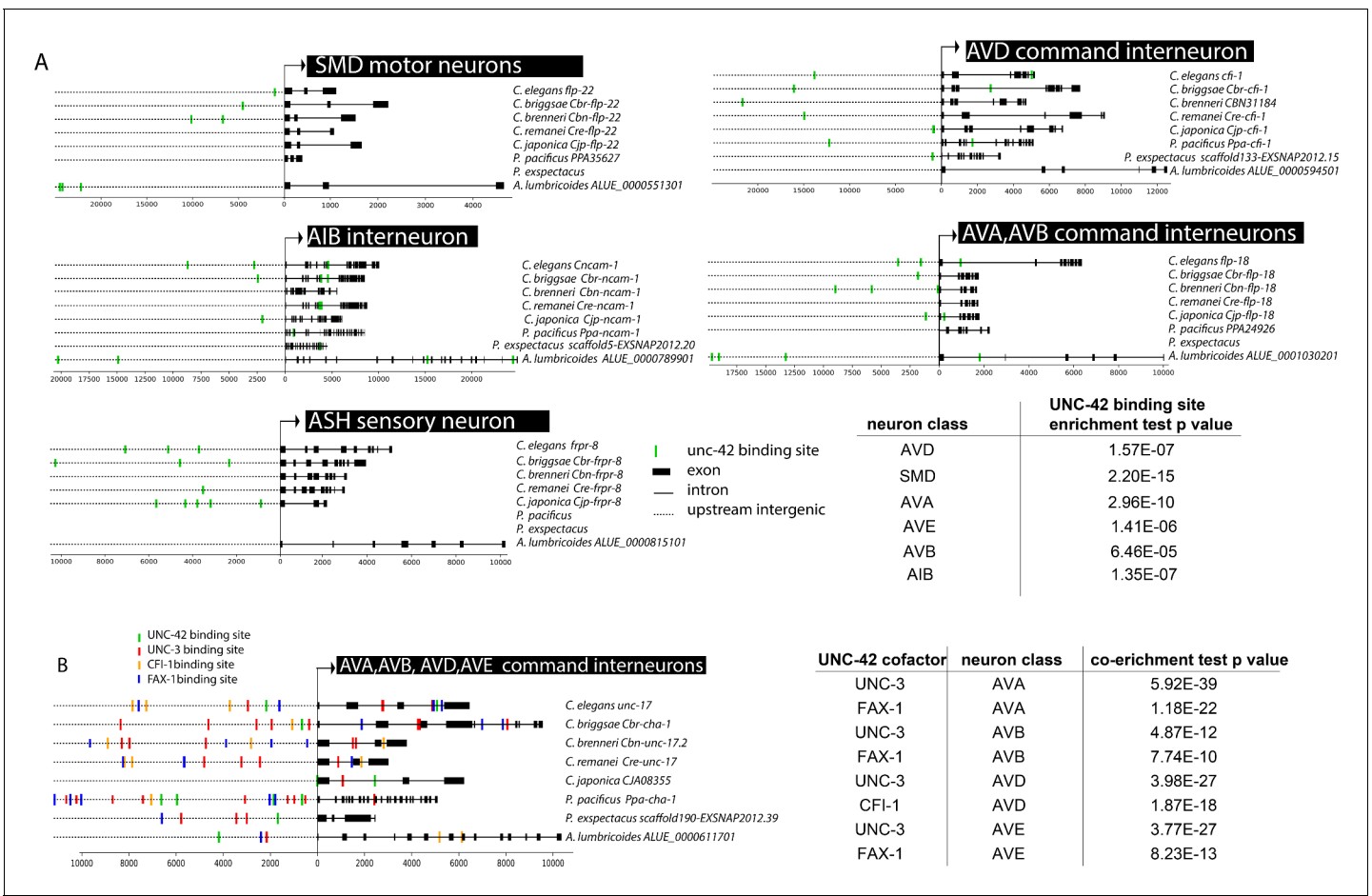

**Figure 12.** *unc-42* cooperates with cofactors in distinct neuron types on the level of target gene promoters. (**A**) Predicted UNC-42 binding sites among orthologs in eight nematode species in *unc-42* expressing neuron classes. Text on right: species name, ortholog name. Table: UNC-42 binding site enrichment in neuron class reporter genes compared to genome-wide binding site data. p-values are from the hypergeometric test for enrichment. (**B**) Predicted *unc-42* cofactor binding sites among *unc-17* orthologs in eight nematode species. Table: co-enrichment of UNC-42 cofactor binding sites in neuron class reporter genes from hypergeometric test comparing genome-wide cofactor binding data.

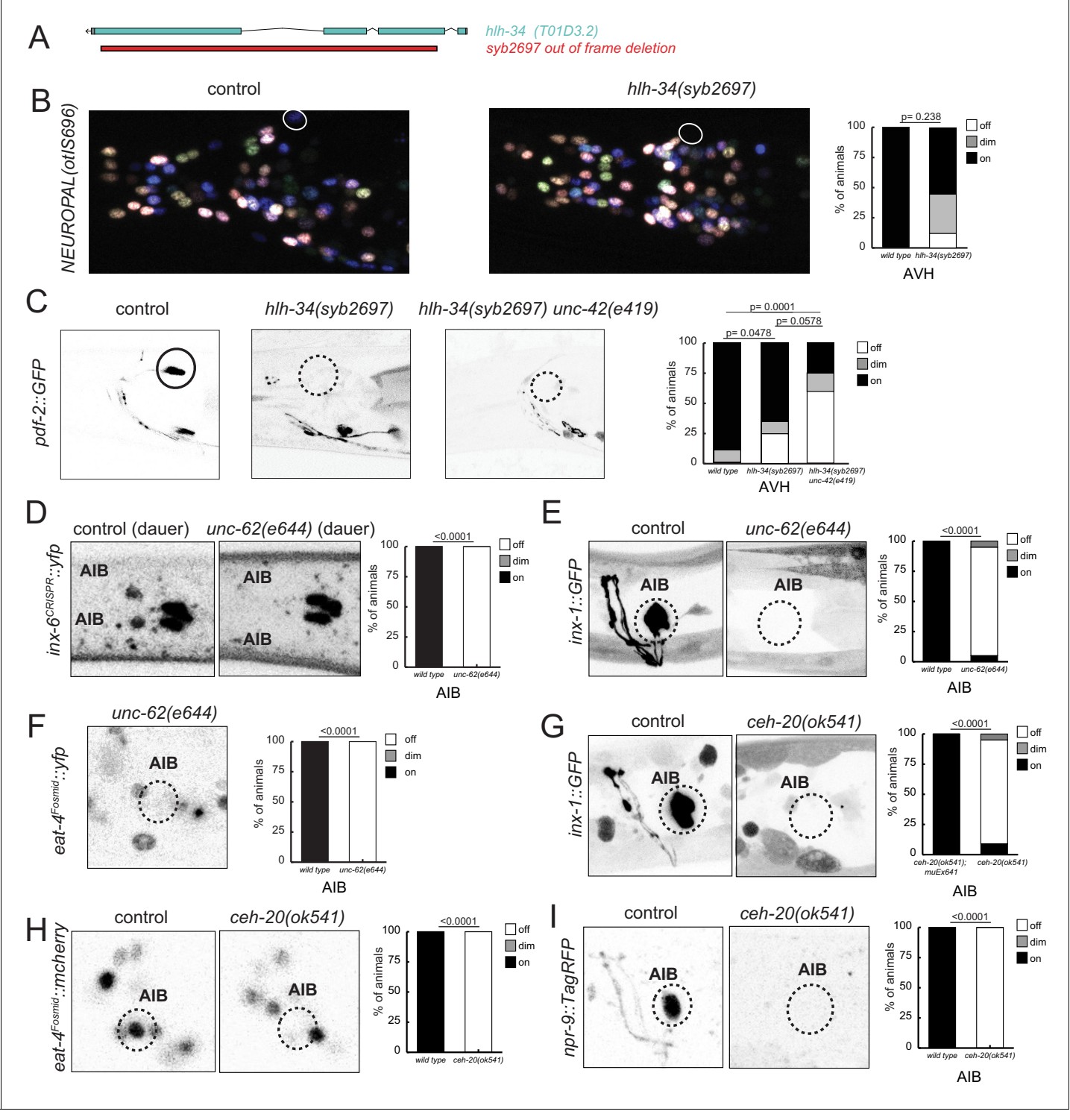

**Figure 13.** *hlh-34, ceh-20/Pbx,* and *unc-62/Meis* are collaborators of *unc-42.* (**A**) The loss-of-function *syb2697* allele of *hlh-34* is a 410 bp deletion. (**B**) *hlh-34*(*syb2697*) mutant animals show an occasional loss of *flp-26::BFP* expression in the AVH neuron in a NeuroPAL(*otIs696*) background. (**C**) *hlh-34* (*syb2697*) mutant animals show an occasional loss of *pdf-2::GFP* expression in the AVH neuron, while *hlh-34*(*syb2697*) *unc-42*(*e419*) double mutants lose or show dim expression in most animals. (**D**) Expression of an *inx-6* reporter allele, which is expressed in AIB neurons at the dauer stage, is lost in *unc-62*(*e644*) mutant dauer animals. (**E**) Expression of an *inx-1* reporter transgene is lost in *unc-62*(*e644*) mutant animals. (**F**) Expression of *eat-4*/VGLUT reporter transgene is lost in *unc-62*(*e644*) mutant animals. (**G**) Expression of an *inx-1* reporter transgene is lost in *ceh-20*(*ok541*) mutant animals (scored as arrested larvae). (**H**) Expression of *eat-4*/VGLUT reporter transgene is lost in *ceh-20*(*ok541*) mutant animals (scored as arrested larvae). (**I**) Expression

*Figure 13 continued on next page*

*Figure 13 continued*

of a neuropeptide receptor, *npr-9* reporter transgene is lost in *ceh-20*(*ok541*) mutant animals (scored as arrested larvae). p-values shown by Fisher's exact test.

The online version of this article includes the following figure supplement(s) for figure 13:

**Figure supplement 1.** *ceh-24* affects SMB motor neuron, not SMD motor neuron differentiation.

**Figure supplement 2.** *unc-42* controls expression of collaborating transcription factors.

## Discussion

Most transcription factors are employed in distinct cell types to exert distinct, cell type-specific functions. *unc-42* is an example of a relatively widely expressed transcription factor, operating in 15 (of the 118) distinct neuron classes of *C. elegans*. Taking a whole animal perspective, we carved out three common themes in the function of *unc-42* in all these different cell types.

First, from the perspective of individual *unc-42(+)* neurons, *unc-42* appears to act as a terminal selector of neuronal identity in each neuron type it is expressed in. *unc-42* is not required for neuron generation or adoption of panneuronal features, but it initiates their respective terminal differentiation program, as inferred by the requirement of *unc-42* for the expression of a host of terminal marker genes of the respective neuron classes. Not every single identity marker is completely affected in *unc-42* mutants, likely due to partial compensation by other cooperating terminal selectors. Based on its continuous expression throughout larval and adult stages, UNC-42 likely also maintains the expression of its effector genes, as demonstrated for other terminal selector-type transcription factors (*Leyva-Díaz and Hobert, 2019*). Consequently, *unc-42* mutants display locomotory defects that phenocopy defects observed upon surgical removal of individual *unc-42(+)* neurons. Moreover, removal of individual *unc-42*-target genes phenocopies behavioral defects observed upon loss of *unc-42*, further corroborating the relevance of *unc-42* and its targets in controlling neuron function. Based on our binding sites analysis, we furthermore predict that UNC-42 directly controls the expression of terminal effector genes in different neuron types. The identification of *unc-42* as a terminal selector in many different neuron types also corroborates the importance of homeobox genes in neuronal identity specification, inferred from past studies of homeobox gene expression and function in *C. elegans* (*Hobert, 2016*; *Reilly et al., 2020*).

The second common theme of UNC-42 function lies in its ability to interact with different collaborating terminal selectors in different neuron types to specify the expression of distinct downstream target genes. These include the *unc-3* transcription factor in command interneurons, the *ceh-24* or *lim-4* transcription factors in neck motor neurons, or the *fax-1* transcription factors in the peptidergic AVK interneuron (*Figure 14A*). Such neuron-type-specific cohorts of collaborating terminal selectors have been termed 'core regulatory complexes' (*Arendt et al., 2016*).

The examination of *unc-42* function in many different neuron types, as well as the genetic interactions with collaborating terminal selectors, adds a number of important nuances to the terminal selector concept: in the *unc-42* single mutant, both the penetrance and expressivity of effects on terminal marker of neuronal identity vary from target gene to target gene and from cell to cell and even vary on the same target gene in different cells. In several cases, we have explicitly shown that mild effects in the *unc-42* single mutants can be enhanced by removing a co-terminal selector. In other cases, there is no room for such enhancement because defects are already fully penetrant in the *unc-42* single mutant. These cell- and target gene-specific effects of a terminal selector are likely a reflection of the distinct mechanisms by which transcription factors activate their targets. In those cases where UNC-42 binds to its target with a cofactor in a strictly cooperative manner, removal of either UNC-42 or its cofactor(s) is expected to result in fully penetrant and expressive defects. A precedent for such scenario is the TTX-3/CEH-10 heterodimer that cooperative binds to *cis*-regulatory motifs present in AIY neuron-expressed genes (*Wenick and Hobert, 2004*). In contrast, UNC-42 may also interact with other target genes in the context of a 'transcription factor collective' (*Junion et al., 2012*), in which transcription factors bind separately to their target promoter and the loss of individual components of the collective can be partially compensated for by other transcription factors in the collective. One example for this model of terminal selector function in *C. elegans* neurons is observed in dopaminergic neurons (*Doitsidou et al., 2013*). Based on the

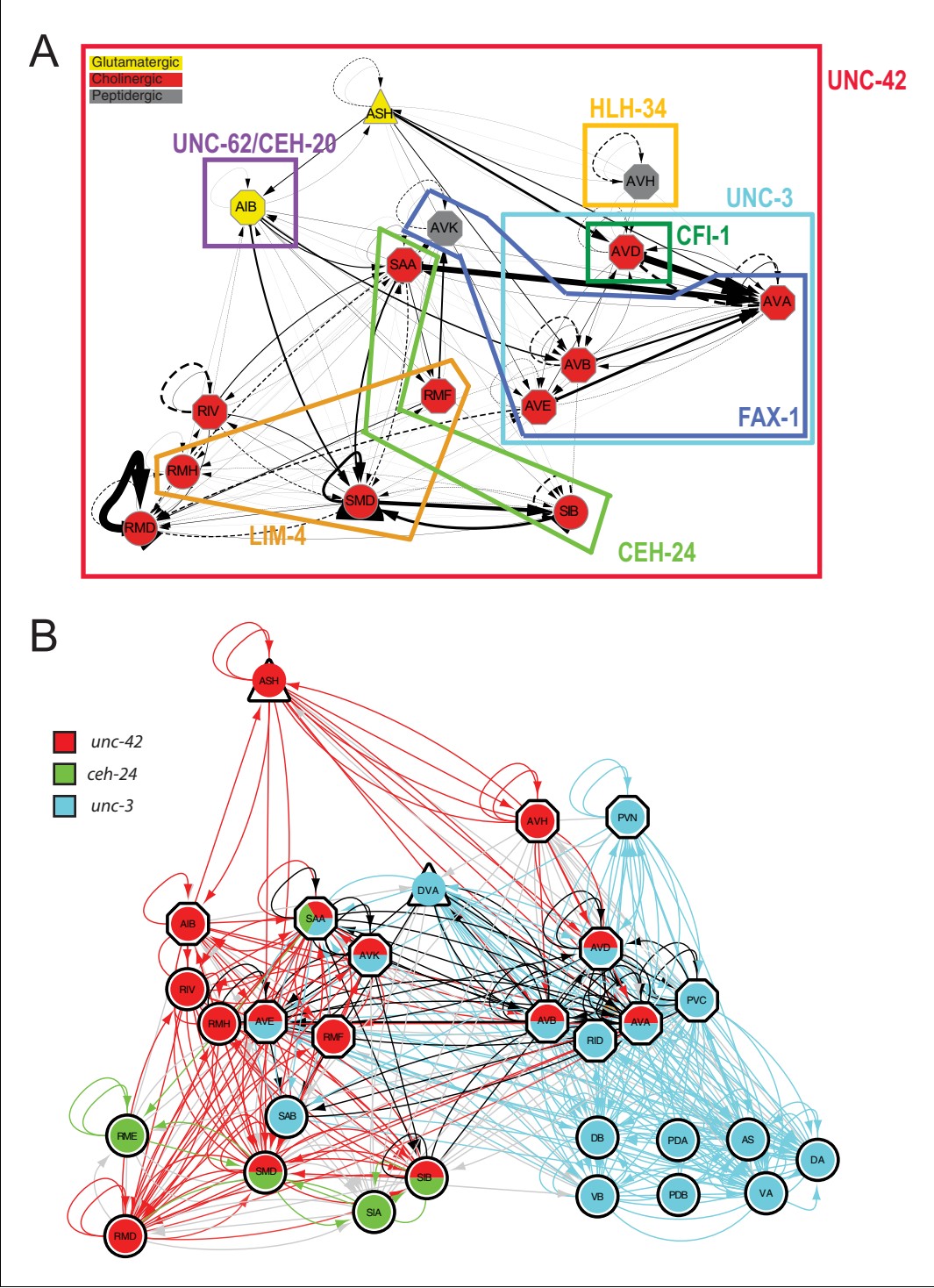

**Figure 14.** Overlapping circuit organizers may assemble individual circuits into larger-scale connectomes. (A) Summary of cofactors for UNC-42. Each colored box indicates where the respective transcription factor is expressed within the set of UNC-42(+) neurons and required, like *unc-42*, for its proper specification. The function of *lim-4* in the SMD neuron is discussed in the accompanying manuscript by Glenwinkel et al. *lim-4* is also expressed in SIB, where its function has not yet been examined. In addition to the factors shown here, each neuron class shown here expresses a unique combination of homeobox genes (*Reilly et al., 2020*), which are candidates to be additional cofactors of *unc-42*. (B) Circuit diagram showing neurons that express *unc-42*, *unc-3*, and *ceh-24* and require these factors for their identity specification. Nodes are colored to illustrate transcription factor expression. Edges are colored if both the source and target neurons express the respective transcription

*Figure 14 continued on next page*

*Figure 14 continued*

factor. Edges are colored in black if more than one transcription factor is expressed in both the source and target neurons. The display is by Cytoscape (https://cytoscape.org/). Nodes are arranged hierarchically, as described (*Cook et al., 2019*). *lim-4*-expressing neurons are not shown here, in part because the function of *lim-4* is not currently known for all *lim-4*-expressing neurons, but also to not further complicate the diagram.

phenotypic analysis described here, we envision that UNC-42 acts in this manner in the majority of neuronal cell types.

The third common theme that emerges from our nervous system-wide analysis of *unc-42* function is that all *unc-42(+)* neurons are synaptically interconnected. This suggests that *unc-42* may also have a role in assembling neurons into functional circuitry. Interestingly, the processes of *unc-42(+)* neurons traverse distinct neighborhoods ('strata') of the nerve ring (*Moyle et al., 2021*; *Brittin et al., 2021*), indicating that *unc-42* may facilitate intra-strata connectivity, thereby ensuring the coordination of information flow through distinct neighborhoods. We provide evidence for a role of *unc-42* in indeed defining synaptic connectivity by demonstrating synaptic connectivity defects in *unc-42* mutants, as inferred by an ultrastructural analysis, but also by the loss of expression of genes involved in electrical synapse formation (innexins) as well as loss of genes possibly involved in synaptic targeting/synapse formation. Taken together with the effect of *unc-42* on neurotransmitter, neurotransmitter receptor, and neuropeptide expression, *unc-42* therefore coordinates both circuit assembly and signaling within this circuit. Whether ectopic misexpression of *unc-42* is sufficient to recruit such ectopic neurons into the set of *unc-42(+)* interconnected neurons is an obvious next question, but will require an improvement in available tools to visualize synaptic wiring.

A detailed comparison of expression patterns of transcription factors and synaptic connectivity reveals several transcription factors whose expression is, like *unc-42*, enriched in synaptically connected neurons. Many of these transcription factors have been shown to be required to control the identity of individual neurons that express these transcription factors, that is, they act as terminal selectors that coordinate the expression of many/most/all terminal identity features of a neuron. Of particular note is the observation that several of the transcription factors that cooperate with *unc-42* in subsets of *unc-42(+)* neurons to specify the identity of specific neurons (including *ceh-24*, *lim-4*, and *unc-3*) are also expressed in synaptically connected sets of neurons. Hence, one can envision that complex circuitry is defined by nest, partially overlapping sets of terminal selector-type transcription factors, each of which define the assembly of groups of neurons into synaptic pathways (*Figure 14B*). For example, we had previously already noted that the *unc-3* terminal selector is expressed in and functions to specify synaptically connected neurons, namely head and tail command interneurons and ventral nerve cord motor neurons, which are innervated by these head and tail commend neurons (*Pereira et al., 2015*). *unc-3* and *unc-42* expression and function overlap in head command interneurons. Hence, a synaptic pathway from sensory input by the polymodal ASH neuron to motor neuron innervation of body wall muscle can be defined by the integration of two sets of synaptically interconnected neurons that are specified by two transcription factors. Similarly, synaptic pathways from sensory to various head motor neurons, and hence, head muscle are defined by overlapping sets of *unc-42(+)* and *ceh-24(+)* neurons (*Figure 14B*).

The association of transcription factors with synaptic connectivity appears to be evident in other nervous systems as well, from other invertebrates to the vertebrate central nervous system. In the perhaps simplest example, motor neuron innervation and their target tissue have been found to rely on matching HOX cluster gene expression (*Arenkiel et al., 2004*; *Hessinger et al., 2017*). Similarly, matching Hoxc8 gene expression specifies motor-sensory neuron connectivity in proprioceptive circuits of the mouse spinal cord (*Shin et al., 2020*) and Shox2 gene specifies interconnected neuron types in the spinal cord (*Ha and Dougherty, 2018*). In the central brain, the Otx2 homeobox gene was found to define and specify neurons in a subcircuit of the habenulo-interpeduncular system (*Ruiz-Reig et al., 2019*) and the homeobox Dbx1 specifies functionally interconnected neurons in the hypothalamus (*Sokolowski et al., 2015*). Perhaps the most striking example is the Phox2 gene, which controls the differentiation of a class of interconnected neurons that form a sensory reflex circuit in the autonomous nervous system (*Dauger et al., 2003*). Another very specialized example can be found in the nervous system of *Drosophila* males, where the transcription factor Fruitless is thought to assemble neurons into functional circuitry (*Stockinger et al., 2005*). It is remarkable that,

with the exception of Fruitless, all the above-mentioned factors are homeobox genes, like *unc-42*, which is in line with their striking predominance in neuronal identity control (*Reilly et al., 2020*).

One way to think about such circuit association is in the context of evolution of neuronal circuitry. Perhaps *unc-42* initially specified the identity of a group of very similar, if not identical, neurons and specified their interconnectivity via control of a homophilic synaptic adhesion molecule. *unc-42* may then have started to collaborate with other transcription factors that were expressed only in a subset of these interconnected neurons to make these neurons become more and more different from one another, but still retaining their interconnectivity. Alternatively, through the gain of UNC-42 expression a neuron previously utilized in one set of interconnected neurons may now become connected to other UNC-42(+) neurons, thereby wiring together originally distinct synaptic pathways. It will be fascinating to assess whether such mechanisms of circuit evolution can be inferred from examining terminal selector expression and synaptic wiring in distantly related nematode species.

## Materials and methods

### *C. elegans* mutant strains and transgenes

*C. elegans* strains used in this study are listed in *Supplementary file 5*. The wild type strain was Bristol N2. Worms were grown on nematode growth media (NGM) agar plates seeded with bacteria (OP50) as a food source.

All transgenes are referenced in *Supplementary file 5*. Contrary to a previous report (*Cunningham et al., 2012*), *hlh-34* transgene reporters are expressed exclusively in AVH, not in AVJ. This cell identification was done with specific landmark strains and will be reported elsewhere. It is also consistent with recent scRNA data, which revealed *hlh-34* expression exclusively in AVH (*Taylor et al., 2021*).

### Light microscopy

*C. elegans* were anesthetized using 100 mM sodium azide and placed on 5% agar pads on glass slides. All images were acquired using a Zeiss 880 laser-scanning confocal. Z-stack images (each ~0.5 µm thick) were acquired using the Zen software and analyzed using the Zen software or ImageJ. Representative images are shown following orthogonal maximum intensity projection of 2–25 z-stacks.

### Neuron identification

Reporter expression analysis was determined by confocal microscopy. Cell identification was done by assessing position and size using Nomarski optics and by crossing with neuronal landmark reporter strains *eat-4 (otIs518, otIs388)* (*Serrano-Saiz et al., 2013*), *cho-1 (otIs544, otIs354)* (*Pereira et al., 2015*), and *NeuroPAL (otIs696)* (*Yemini et al., 2021*). Expression in a subset of sensory neurons was confirmed by dye filling with DiD.

### Quantification of neuroanatomical features

For quantification of axon length shown in *Figure 8*, confocal Z-stacks were opened using FIJI software and were loaded into the Simple Neurite Tracer plugin. Using this plugin, the axon emerging from the soma of AVH was traced across multiple Z-stacks and summed to calculate total axon length.

For quantification of RAB-3::GFP puncta shown in *Figure 8—figure supplement 3*, manual counting was performed using the ZEN software.

For quantification of cell nucleus positions, confocal Z-stacks were opened in the NeuroPAL ID software as described in *Yemini et al., 2021*. Cell nuclei were manually identified and aligned X, Y, and Z nucleus positions were assessed by measuring the distance from the point of origin, as determined by the NeuroPAL ID software. The X, Y, and Z nucleus positions of *unc-42* mutant animals were evaluated individually and were not different as compared to wild type animals (data not shown). Additionally, a Euclidean distance was assessed using the distance formula, as shown in *Figure 8—figure supplement 1*.

## Electron microscopy and serial reconstruction

*unc-42(e270)* was fixed as previously described (*White et al., 1986*). These fixed worms were then cut into 50 nm sections using RMC Powertome XL and collected onto grids. The nerve ring region of *unc-42(e270)* was then imaged either manually with a Phillips CM10 TEM or automatically with a JEOL 1400Plus TEM and the SerialEM software. Sections were then aligned and montaged, all of the axons in the nerve ring were serially traced, and synapses were annotated using the TrakEM2 software (*Cardona et al., 2012*). The region imaged, reconstructed, and annotated was ~15 µm in length and included 309 serial sections. Neurons were identified by characteristic synaptic and/or morphological features together with relative cell body position (*Supplementary file 6*). Synapse counts and axon adjacency counts were then extracted using scripts kindly provided by Christopher Brittin (*Brittin et al., 2018*). To compare to the *unc-42(e270)* synapse and axon adjacency counts to the previously described *wild type (N2U)* (*Cook et al., 2019*), the sections were aligned from the beginning of the RMEV neuron nucleus, a neuron that is easily identifiable based on morphology and position, to the anterior end of the nerve ring.

## WormTracker assays

WormTracker assays were conducted and analyzed as previously described (*Yemini et al., 2013*). To avoid any potential variability due to room conditions, mutant and wild type strains were recorded simultaneously for each experiment. Briefly, individual L4 worms were placed on unseeded NGM plates. These worms were then tracked for 3 min with the WormTracker 2.0 (WT2) software, which tracks and records each worm with a camera.

## Statistical analysis of *unc-42* mutant phenotypes

For the categorical data shown in *Figures 3–6*, *8*, *10*, *11,* and *13*, statistical analysis was performed using Fisher's exact test. For the numerical data shown in *Figures 2*, *7,* and *8,*, statistical analysis was performed using a one-way ANOVA followed by a post-hoc Tukey HSD test. Where appropriate, p-values were adjusted using a FDR correction for multiple testing.

## Correlating gene expression with synaptic connectivity

The probability that a transcription factor is expressed in a set of neurons that are more interconnected than the whole connectome was calculated with a probability mass function using a binomial distribution. Analysis was performed as described (*Arnatkevičiūtė et al., 2018*) with modifications, using an updated transcription factor expression database. Connectivity data was taken from https://www.wormwiring.org and described in *Cook et al., 2019*.

The probability of having *k* success in *n* trials is given by the probability mass function:

$$\Pr(k;n,p) = \Pr(X = k) = \binom{n}{k} p^k (1-p)^{n-k}$$

We define the 'probability of success' (p) as the probability that any two given neurons are connected in the somatic (i.e., non-pharyngeal) *C. elegans* hermaphrodite connectome, excluding the pharyngeal neurons. This was calculated by examining all possible neuron pairs, excluding interclass pairs, where order matters (e.g., A–B is not the same as B–A), and totaling how many of these pairs were connected by either a chemical and/or electrical connection. Both electrical and chemical synapses were doubly counted (e.g., A > B and A < B were counted as two connections).

The 'number of trials' (n) was determined by totaling all possible pairs of neurons in which the transcription factor was expressed, excluding interclass pairs, where order matters.

The 'number of successes' (k) was determined for each transcription factor by totaling how many of these pairs were chemically and/or electrically connected.

A probability mass function calculation using a binomial distribution was then performed for each transcription factor, and a p-value was calculated. These p-values were then corrected for multiple testing using a FDR correction. Custom computation scripts are available at https://github.com/hobertlab/Berghoff_2021 (*Berghoff and Hobert, 2021a*, copy archived at swh:1:rev:2e64fea4812ce726f3e679dca3f691d3e866af43 *Berghoff and Hobert, 2021b*).

## TargetOrtho analysis

Transcription factor DNA binding motifs from the CISBP version 2.0 database (*Weirauch et al., 2014*) (unc-42: M03874_2.00,cfi-1:M01667_2.00,fax-1:M06432_2.00) and unc-3 (*Kratsios et al., 2011*) were used with TargetOrtho2.0 (*Glenwinkel et al., 2014*; Glenwinkel et al., unpublished) (FIMO p value threshold: 1e-4) to identify binding sites among orthologous coding gene loci in eight nematode species. Binding site enrichment tests were conducted using Python's hypergeom function among *C. elegans* neuron class reporter genes. Reporter genes per neuron class are from the Hobert lab's curated and recently updated collection of reporter genes (see Brain Atlas in *Hobert et al., 2016*). *C. elegans* coding gene annotations are from Wormbase version WS264. UNC-42 binding site enrichment tests: the expected proportion of binding sites is computed as the number of coding genes in the genome with at least one binding site divided by the total number of coding genes annotated. The observed proportion of binding sites is the number of neuron class-specific reporter genes that have at least one binding site in upstream intergenic or intronic regions. UNC-42 cofactor binding site enrichment tests: the expected proportion of coding genes with cofactor binding sites was computed by multiplying together the proportion of genes in the whole genome with at least one binding site match for both cofactors examined. The observed proportion is the number of neuron class reporter genes with at least one binding site from each cofactor.

## Acknowledgements

We thank Nichol Thomson for fixing *unc-42* mutant animals; Chi Chen for generating transgenic lines; Isabel Beets for sharing unpublished data; and Paschalis Kratsios, Nuria Flames, Austen Sitko, and Lisa Goodrich for comments on the manuscript. This work was funded by NIH OD010943 (to DHH), National Science Foundation (grant 1351649 to DMF) and NIH 1R01NS110391 (to OH), and the HHMI.

## Additional information

### Competing interests

Oliver Hobert: Reviewing editor, *eLife*. The other authors declare that no competing interests exist.

### Funding

| Funder | Grant reference number | Author |
|---|---|---|
| Howard Hughes Medical Institute | | Oliver Hobert |
| National Institutes of Health | OD010943 | David H Hall |
| National Science Foundation | 1351649 | Denise M Ferkey |
| National Institutes of Health | 1R01NS110391 | Oliver Hobert |

The funders had no role in study design, data collection and interpretation, or the decision to submit the work for publication.

### Author contributions

Emily G Berghoff, Conceptualization, Formal analysis, Investigation, Visualization, Writing - review and editing; Lori Glenwinkel, Abhishek Bhattacharya, HaoSheng Sun, Formal analysis, Investigation, Writing - review and editing; Erdem Varol, Formal analysis, Investigation, Methodology; Nicki Mohammadi, Amelia Antone, Yi Feng, Ken Nguyen, Steven J Cook, Jordan F Wood, Neda Masoudi, Cyril C Cros, Yasmin H Ramadan, Formal analysis, Investigation; Denise M Ferkey, Supervision, Project administration, Writing - review and editing; David H Hall, Supervision, Investigation, Project administration, Writing - review and editing; Oliver Hobert, Conceptualization, Supervision, Funding acquisition, Writing - original draft, Project administration

## Author ORCIDs
HaoSheng Sun (iD) http://orcid.org/0000-0003-3919-559X
Steven J Cook (iD) http://orcid.org/0000-0002-1345-7566
David H Hall (iD) http://orcid.org/0000-0001-8459-9820
Oliver Hobert (iD) https://orcid.org/0000-0002-7634-2854

## Decision letter and Author response
Decision letter https://doi.org/10.7554/eLife.64903.sa1
Author response https://doi.org/10.7554/eLife.64903.sa2

## Additional files

### Supplementary files
• Supplementary file 1. UNC-42(+) neurons and their function.

• Supplementary file 2. Strata placement of UNC-42(+) neurons. Clustering outputs of *Brittin et al., 2021*, *Moyle et al., 2021* are shown for each neuron. Discordant clustering results of neuronal sub-classes are shown for RMD and SIB.

• Supplementary file 3. Homeobox gene expression correlating with synaptic connectivity. See Materials and methods for details of this analysis. p-values were calculated using the binomial distribution probability mass function and were adjusted for multiple testing using a false discovery rate correction (see Materials and methods). All genes passing the p<0.05 significance threshold are shown here. Overlap with network differential gene expression analysis shaded in yellow. Red font: genes shown to be involved in neuronal identity regulation (*Reilly et al., 2020*; *Hobert, 2016*).

• Supplementary file 4. List of motion behaviors examined in wild type and *unc-42(e419)* animals. Green indicates motion features that are not significantly different, while red indicates motion features that are significantly different (p<0.05) between wild type and *unc-42(e419)* animals. Motion features were measured for the entire animal, and in the head, tail, and midbody regions. They were measured when the animal was moving forward, backward, or paused. Features were measured accounting for when the data is signed, by absolute data values ('absolute'), positive data values only ('positive'), and negative data values only ('negative'). Motion features are described by the frequency, the time spent, and the distance covered. The animal's velocity is described in two parts: speed and motion direction. Crawling, an undulation of the animal's body used for movement, is described as an amplitude and a frequency. Foraging, a rapid movement of the nose as the animal explores its environment, is described as an amplitude and a speed. An omega turn is when the animal bends sharply such that the head touches the tail in order to reverse direction. An upsilon turn is when the animal bends shallowly in order to reverse direction. Time ratio is defined as the total time spent in a particular behavior divided by the total time. See *Yemini et al., 2013* for more detailed feature descriptions.

• Supplementary file 5. Strains used in this study.

• Supplementary file 6. Methods for neuron identification in electron micrographs of *unc-42(e270)*.

• Transparent reporting form

## Data availability
All data generated or analysed during this study are included in the manuscript and supporting files.

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
