## [Decision Letter]

**Acceptance summary:**

The authors explore how neural circuits are established in *C. elegans*. They identify a transcription factor present in 15 neuronal classes that are all synaptically connected, and show that this factor UNC-42/Prop-1 coordinately regulates both differentiation and connectivity of these neurons. Major strengths are the comprehensive analysis, high quality of the data, and the power of the concept (one factor – one circuit) to guide further research in all organisms.

**Decision letter after peer review:**

Thank you for submitting your article "The Prop1-like homeobox gene unc-42 specifies the identity of synaptically connected neurons" for consideration by *eLife*. Your article has been reviewed by 2 peer reviewers, onw of whom is a member of our Board of Reviewing Editors, and the evaluation has been overseen by Piali Sengupta as the Senior Editor. The reviewers have opted to remain anonymous.

The reviewers have discussed the reviews with one another and the Reviewing Editor has drafted this decision to help you prepare a revised submission.

We would like to draw your attention to changes in our policy on revisions we have made in response to COVID-19 (https://elifesciences.org/articles/57162). Specifically, when editors judge that a submitted work as a whole belongs in *eLife* but that some conclusions require a modest amount of additional new data, as they do with your paper, we are asking that the manuscript be revised to either limit claims to those supported by data in hand, or to explicitly state that the relevant conclusions require additional supporting data.

Summary:

The authors explore how neural circuits are established in *C. elegans*. They identify a transcription factor, Prop1/Unc-42, present in 15 neuronal classes that are all synaptically connected, and show that this factor coordinately regulates both differentiation and connectivity of these neurons. Analysis of the expression pattern of many other transcription factors reveals a striking relationship between transcription factors and neuronal network. Weaknesses are the minimal statistics showing preferential connectivity, and the functional analysis of connectivity. Strengths are the comprehensive analysis, high quality of the data, and the power of the concept (one factor – one circuit) to guide further research in all organisms.

Essential revisions:

1. The analysis of connectivity within the 15 classes is weak. The UNC-42 neurons are in the head, but comparing them to random neurons throughout the body is not a fair test: the control group should be a similar number of UNC-42-negative neurons in the same region of the head. Data on connectivity from UNC-42+ to UNC-42- neurons should be given; currently it is impossible to tell if there is more connectivity within the group or from the group to the strongest connected UNC-42-negative neurons (or from UNC-42-negative neurons to UNC-42+ neurons). This is the major conclusion of the paper and needs additional analysis to make the conclusions more convincing.

2. The behavior should be clarified. It appears unc-42 mutants fail all motor tasks. If they are specifically poor in nociceptive escape tasks that should be made clear. Documenting behavioral specificity would help support the specificity of the UNC-42 circuit. That is, does UNC-42 generate a specific behavior (via its downstream circuit) or does UNC-42 simply eliminate motor function?

3. Figure 3: There is considerable variation in penetrance and expressivity between different neuron types regarding the phenotype (loss of cholinergic and glutamatergic transmitter markers) of unc-42 mutants. Yet the cartoon displays these effects as complete, which would seem to be an exaggeration. Also, why would there be such variability in the phenotype between cell types? Wouldn't "master terminal selectors" show complete penetrance and expressivity?

4. Do UNC-42+ neurons share a common neuropil target? Do they share a common pathfinding intermediate (e.g. ipsi vs contralateral)? Common targeting features would help support a common molecular pathway with UNC-42 at the top. The opposite is interesting too – perhaps UNC-42 regulates different pathfinding gene batteries just like it regulates different neurotransmitter biosynthetic enzymes – but it would be good for the reader to be given this information.

5. It seems unfair to attribute whole animal mutant phenotypes (Figure 7B) to the UNC-42 circuit. Or am I missing something: were these neuron-specific experiments? If the UNC-42 phenotype is severe (all motor behavior affected) then how relevant are phenocopies? Anything to strengthen this section would help the paper.

6. It seems the TEM analysis – impressive though it is – is not properly analyzed or discussed. It is true that the SAAVL > AVAL data strongly support the idea that proximity is not sufficient for connectivity, but some of the neurons show different results, such as increased connectivity in the mutant, rather than decreased. This needs to be noted in the paper. Also, it could be discussed in the context of synaptic choice in *Drosophila* lamina neurons, where loss of the CAM dip-beta does not remove synapses, but rather it is required to restrict subcellular localization (thus mutants have ectopic synapses) (Xu /Pecot, Neuron 2019).

[Editors' note: further revisions were suggested prior to acceptance, as described below.]

Thank you for submitting your article "The Prop1-like homeobox gene unc-42 specifies the identity of synaptically connected neurons" for consideration by *eLife*. Your article has been reviewed by 3 peer reviewers, including Chris Q Doe as the Reviewing Editor and Reviewer #1, and the evaluation has been overseen by Piali Sengupta as the Senior Editor.

Essential revisions:

These are all text or figure changes. No experiments are required. Yet the comments are important to improve the clarity and readability of the paper.

1. Line 139 says (REFs). Need to add the references.

2. Line 173-175 says 17 additional HD transcription factors (in addition to Unc-42) show common connectivity. This should be documented in the Results section with the names of the TFs and the statistics on connectivity. Perhaps a supplemental table listing all 18 HD TFs that fall in this category. The discussion should be expanded to discuss what these 18 HD TFs have in common that may distinguish them from the other 65 HD TFs that don't have common connectivity.

3. The legend to Figure 1 mentions panel F with NDGE analysis. But there is no panel F in Figure 1. Please add.

4. The authors claim that mild effects (variability/penetrance/expressivity) is common for "terminal selectors", which were originally defined as "controlling all aspects of a specific neuron's cell specific gene expression", "during development and into adults". How mild can the effects be then, to still allow for the "terminal selector" definition to be used for a TF.

And the role of "terminal selectors" is compensated by "cofactors", which are also "terminal selectors"? Or perhaps merely "late acting, partially important, cell fate determinants"?

5. Lines 482-483, 526-545, 562. The original definition of "terminal selectors' was that they should control "all sub-type defining genes". But now the authors claim that this is a "grey zone", and that 19/20 (che-1) or 37/40 (unc-3) targets is sufficient. So what is the percentage of sub-type defining genes that a TF needs to regulate to qualify as a "terminal selector, 95%, 90%?

6. The authors did not disclose the nature of unc-42(e270), which was used in ultrastructural analysis and only one allele unc-42(e419) was used throughout the entire study. Please acknowledge in the paper, or correct the text to show other alleles.

7. Some data (e.g. Figure 7-S1, Figure 8 S1) are presented in a crowded and nearly invisible manner, diminishing the value to any readers.

---

## [Author Response]

Essential revisions:1. The analysis of connectivity within the 15 classes is weak. The UNC-42 neurons are in the head, but comparing them to random neurons throughout the body is not a fair test: the control group should be a similar number of UNC-42-negative neurons in the same region of the head. Data on connectivity from UNC-42+ to UNC-42- neurons should be given; currently it is impossible to tell if there is more connectivity within the group or from the group to the strongest connected UNC-42-negative neurons (or from UNC-42-negative neurons to UNC-42+ neurons). This is the major conclusion of the paper and needs additional analysis to make the conclusions more convincing.

In addition to the analysis that we originally presented (and re-worded for clarity in the revised version), we now added an entirely novel set of statistical analysis that assesses the extent of interconnectivity of unc-42-expressing neuron. This analysis further corroborates that unc-42 indeed defines a set of neurons that are more interconnected than expected by chance and is presented on page 6 of the manuscript. This analysis was conducted by a new author on the manuscript who conducted a similar type of analysis on recently released scRNA data.

2. The behavior should be clarified. It appears unc-42 mutants fail all motor tasks. If they are specifically poor in nociceptive escape tasks that should be made clear. Documenting behavioral specificity would help support the specificity of the UNC-42 circuit. That is, does UNC-42 generate a specific behavior (via its downstream circuit) or does UNC-42 simply eliminate motor function?

We regret that we were not clearer about this. The reviewer would be totally correct in pointing out that loss of the nociceptive escape behavior would have little meaning if unc-42 animals were simply totally immobile, failing all motor tasks. But, no, this is not the case, unc-42 mutants do not simply fail all motor tasks; they are fine in plenty. We now document this properly in a new Supp Table S3 in which we list all locomotory features that we quantified and we refer to this in the main text.

3. Figure 3: There is considerable variation in penetrance and expressivity between different neuron types regarding the phenotype (loss of cholinergic and glutamatergic transmitter markers) of unc-42 mutants. Yet the cartoon displays these effects as complete, which would seem to be an exaggeration. Also, why would there be such variability in the phenotype between cell types? Wouldn't "master terminal selectors" show complete penetrance and expressivity?

We felt that the visualization of partial effects would have made the cartoons (that we use throughout the manuscript) too complex. Note that in many of the cartoon we also need to use multiple color. Introducing an additional level of information for partial effects (eg shading or stippling) would beat the purpose of the schematic. We now indicate in the figure legends that a loss of color indicates the existence of an effect on marker gene expression, irrespective of whether the effect is fully penetrant or not.

In terms of variability/penetrance/expressivity – the literature shows this to be a common feature of loss of transcription factors, including terminal selectors. This partial penetrance has, in a great number of cases, shown to be due to cofactors that can partially compensate for loss of the TF. Such cofactors are different in different cell types and, hence, the extent of the phenotype may differ from cell type to cell type. In the original version of the manuscript, we had only considered a few co-factors, but in the revised version, we have significantly expanded this cofactor theme: While in the original version we had only shown synergistic effects of *unc-42* and *unc-3* in command interneurons (i.e. mild effects in single, enhanced in double), we now similar synergistic effects in three other neuron types, with yet different cofactors: with *hlh-34* in AVH and with *lim-4* in RMF and RMH. The data is shown in Figure 3 (for lim-4) and Figure 13 (for hlh-14). We have updated the summary figure 14 as well.

4. Do UNC-42+ neurons share a common neuropil target? Do they share a common pathfinding intermediate (e.g. ipsi vs contralateral)? Common targeting features would help support a common molecular pathway with UNC-42 at the top. The opposite is interesting too – perhaps UNC-42 regulates different pathfinding gene batteries just like it regulates different neurotransmitter biosynthetic enzymes – but it would be good for the reader to be given this information.

Yes, same neuropil target. We now illustrate this in a new 3D rendering, added to Figure 1.

5. It seems unfair to attribute whole animal mutant phenotypes (Figure 7B) to the UNC-42 circuit. Or am I missing something: were these neuron-specific experiments? If the UNC-42 phenotype is severe (all motor behavior affected) then how relevant are phenocopies? Anything to strengthen this section would help the paper.

Yes, this indeed needs clarification. As stated above, unc-42 mutants do not have all motor behaviors affected. And, no, the mutants were not neuron-specific removals. As such the phenocopies come, of course, with caveats. We consider them not proof of direct relationship, but rather a necessary precondition to consider them good candidates for conferring the unc-42 defects. Not more, not less. We now clarify this in the text (p.15).

6. It seems the TEM analysis – impressive though it is – is not properly analyzed or discussed. It is true that the SAAVL > AVAL data strongly support the idea that proximity is not sufficient for connectivity, but some of the neurons show different results, such as increased connectivity in the mutant, rather than decreased. This needs to be noted in the paper. Also, it could be discussed in the context of synaptic choice in *Drosophila* lamina neurons, where loss of the CAM dip-beta does not remove synapses, but rather it is required to restrict subcellular localization (thus mutants have ectopic synapses) (Xu /Pecot, Neuron 2019).

We are a little reluctant to highlight the relatively modest increases in some connection. In contrast to some of the losses, the increases seem to be in the range of ~2fold, which is more in line with the normal animal to animal variability that one observes. It’s totally possible that this is real, but we do not want to sound like we try to overinterpret our data.

[Editors' note: further revisions were suggested prior to acceptance, as described below.]

Essential revisions:These are all text or figure changes. No experiments are required. Yet the comments are important to improve the clarity and readability of the paper.1. Line 139 says (REFs). Need to add the references.

Fixed.

2. Line 173-175 says 17 additional HD TFs (in addition to Unc-42) show common connectivity. This should be documented in the Results section with the names of the TFs and the statistics on connectivity. Perhaps a supplemental table listing all 18 HD TFs that fall in this category. The discussion should be expanded to discuss what these 18 HD TFs have in common that may distinguish them from the other 65 HD TFs that don't have common connectivity.

Apologies for this oversight. It’s now in Figure 1F and in a new Supplementary File 2.

We actually cut this number down to 8 by introducing the criterion that BOTH statistical methods that use have to show enrichment (before we only used either one). It’s more conservative.

In regard to what those genes have in common in comparison to the other TFs, we are at a loss to explain it. Perhaps a reflection of those being the most ancient TFs to have become recruited into this function, but that’s fantasy.

3. The legend to Figure 1 mentions panel F with NDGE analysis. But there is no panel F in Figure 1. Please add.

Apologies, done (as stated above).

4. The authors claim that mild effects (variability/penetrance/expressivity) is common for "terminal selectors", which were originally defined as "controlling all aspects of a specific neuron's cell specific gene expression", "during development and into adults". How mild can the effects be then, to still allow for the "terminal selector" definition to be used for a TF.

The definition of a defect – no matter whether a marker is completely off or off in some animal (partial penetrance) or less strongly expressed (partial expressivity) – is “any defect” matters, as long as it statistically significant of course. The extent of defects differ by factor/target/cell not based on any real conceptual differences, but based on mere enhancer architecture: In those cases where we know that multiple factors bind in a cooperative manner to target DNA, the effect of removal of either factor is completely penetrant and expressive (example: AIY neuron – ttx-3/ceh-10 or touch neurons – unc-86/mec-3). In other case, terminal selectors work more in a “billboard” type of mechanisms, where it is the additive, non-coperative binding of multiple factors that leads to full target gene activation (example: dopamine neurons – ast-1/ceh-43/ceh-20) – in such case, removal of a single factor can be partially compensated for by the other factors.

And the role of "terminal selectors" is compensated by "cofactors", which are also "terminal selectors"? Or perhaps merely "late acting, partially important, cell fate determinants"?

They would also be terminal selectors.

Overall thee are really good points and are glad to have been prompted to clarify this. The above responses are now added into an additional paragraph in the Discussion.

5. Lines 482-483, 526-545, 562. The original definition of "terminal selectors' was that they should control "all sub-type defining genes". But now the authors claim that this is a "grey zone", and that 19/20 (che-1) or 37/40 (unc-3) targets is sufficient. So what is the percentage of sub-type defining genes that a TF needs to regulate to qualify as a "terminal selector, 95%, 90%?

This is a good question and hard to answer definitively. What’s notable is that empirically, there are very few cases that lie in the middle. Either a TF affects are large majority of markers, or it affects very few (we comprehensively have listed this in a WIRES review in 2016). The accompanying paper that analyses the occurance of terminal selector binding sites takes a more quantitative stab at this problem: If there are more binding sites of TF in the complete battery of neuron-type specific genes (scRNA) than expected by chance, then we count it as a terminal selector.

6. The authors did not disclose the nature of unc-42(e270), which was used in ultrastructural analysis and only one allele unc-42(e419) was used throughout the entire study. Please acknowledge in the paper, or correct the text to show other alleles.

Good catch. The preparation for the EM analysis (i.e. fixing and staining) was done on e270, before the molecular identity of unc-42 was known. e270 is a missense mutation in a conserved homeodomain residue, e419 is a premature stop in the homeodomain. Now stated in paper.

7. Some data (e.g. Figure 7-S1, Figure 8 S1) are presented in a crowded and nearly invisible manner, diminishing the value to any readers.

Since those are online-only, they can be blown up. The are in vector-based AI format, so there is no loss of resolution when blown up.